# Bone Marrow Mesenchymal Stem Cells Promote Ovarian Cancer Cell Proliferation via Cytokine Interactions

**DOI:** 10.3390/ijms25126746

**Published:** 2024-06-19

**Authors:** Kai-Hung Wang, Yu-Hsun Chang, Dah-Ching Ding

**Affiliations:** 1Department of Medical Research, Hualien Tzu Chi Hospital, Buddhist Tzu Chi Medical Foundation, Tzu Chi University, Hualien 970, Taiwan; kennyhug0201@gmail.com; 2Department of Pediatrics, Hualien Tzu Chi Hospital, Buddhist Tzu Chi Medical Foundation, Tzu Chi University, Hualien 970, Taiwan; cyh0515@gmail.com; 3Department of Obstetrics and Gynecology, Hualien Tzu Chi Hospital, Buddhist Tzu Chi Medical Foundation, Tzu Chi University, Hualien 970, Taiwan; 4Institute of Medical Sciences, Tzu Chi University, Hualien 970, Taiwan

**Keywords:** ovarian cancer, bone marrow mesenchymal stem cells, cytokines, IL-6, MCP-1

## Abstract

Bone marrow mesenchymal stem cells (BMSCs) are key players in promoting ovarian cancer cell proliferation, orchestrated by the dynamic interplay between cytokines and their interactions with immune cells; however, the intricate crosstalk among BMSCs and cytokines has not yet been elucidated. Here, we aimed to investigate interactions between BMSCs and ovarian cancer cells. We established BMSCs with a characterized morphology, surface marker expression, and tri-lineage differentiation potential. Ovarian cancer cells (SKOV3) cultured with conditioned medium from BMSCs showed increased migration, invasion, and colony formation, indicating the role of the tumor microenvironment in influencing cancer cell behavior. BMSCs promoted SKOV3 tumorigenesis in nonobese diabetic/severe combined immunodeficiency mice, increasing tumor growth. The co-injection of BMSCs increased the phosphorylation of p38 MAPK and GSK-3β in SKOV3 tumors. Co-culturing SKOV3 cells with BMSCs led to an increase in the expression of cytokines, especially MCP-1 and IL-6. These findings highlight the influence of BMSCs on ovarian cancer cell behavior and the potential involvement of specific cytokines in mediating these effects. Understanding these mechanisms will highlight potential therapeutic avenues that may halt ovarian cancer progression.

## 1. Introduction

Ovarian cancer is the leading cause of mortality among gynecologic cancers, which saw 21,750 new cases and 13,490 deaths in the USA in 2020 [1]. In Taiwan, 1677 new patients were diagnosed with ovarian cancer in 2019 (incidence: 9.9/100,000), and 696 deaths occurred in 2021 (according to the Ministry of Health and Welfare, Taiwan). The epithelial type is the most common form of ovarian cancer, which is often diagnosed at advanced stages [2]. Current management includes debulking surgery, adjuvant chemotherapy, and/or targeted therapy [3]. The recurrence rate is 70% within the first five years of diagnosis among advanced-stage patients [3]. Hence, improvement in the treatment of advanced-stage ovarian cancer is urgently required.

The tumor microenvironment (TME) is a complex and progressively evolving environment. It contains fibroblasts, stromal cells, endothelial cells, mesenchymal stem cells (MSCs), and innate and adaptive immune cells [4]. The tumor cells interact with bone marrow mesenchymal stem cells (BMSCs) and their secreted cytokines in the TME [4].

BMSCs are multipotent cells found within the bone marrow that can differentiate into various cell types, including blood cells (such as red blood cells, white blood cells, and platelets) and cells of the mesenchymal lineage (such as osteoblasts, adipocytes, and chondrocytes) [5]. These cells play a crucial role in replenishing and maintaining the body’s blood cell supply through hematopoiesis and in tissue repair and regeneration through their ability to differentiate into specialized cell types [6]. BMSCs have garnered significant attention in research and clinical applications due to their regenerative potential and ability to contribute to treating various diseases and injuries [7]. BMSCs produce cytokines like interleukins (IL-6, IL-8), tumor necrosis factor-alpha (TNF-α), and transforming growth factor-beta (TGF-β), influencing self-renewal, differentiation, and interactions within the bone marrow niche [8]. Additionally, BMSCs respond to cytokines from neighboring cells; inflammatory cytokines stimulate anti-inflammatory responses and tissue repair [9]. Hematopoietic cell-derived cytokines, like granulocyte colony-stimulating factor (G-CSF), also affect BMSC function and hematopoietic stem cell maintenance [10]. This intricate interplay regulates tissue homeostasis, regeneration, and immune modulation.

Within the primary ovarian tumor microenvironment, BMSCs may migrate to the tumor stroma in response to chemotactic signals released by cancer cells or other stromal cells [11]. Once in the tumor vicinity, BMSCs can directly interact with ovarian cancer cells through cell-to-cell contact or paracrine signaling [12,13]. As ovarian cancer progresses, metastases often occur in the peritoneal cavity, including the omentum, peritoneum, and abdominal organs [14]. BMSCs may home in on these metastatic sites and contribute to forming a supportive microenvironment for cancer cell survival and growth [15]. The bone marrow serves as a reservoir for stem and progenitor cells, including BMSCs. The interaction between BMSCs and ovarian cancer cells can occur through various mechanisms, including paracrine signaling, exosome-mediated communication, and direct cell-to-cell contact [16,17,18].

MSCs can engage in self-renewal and tri-lineage differentiation [19], and their anti-inflammatory and immunomodulatory effects are mediated by paracrine cytokines [19]. MSCs influence tumor evolution and advancement by fostering stem-like properties in tumor cells, facilitating migration, stimulating angiogenesis, dampening immune reactions, and instigating drug resistance [20]. MSCs secrete various factors, including prostaglandin E2, indoleamine 2,3-dioxygenase, interferon-gamma, IL-4, IL-6, IL-8, monocyte chemoattractant protein-1 (MCP-1), and TGF-β1, which interact with the immune cells to downregulate the anti-tumor immune response [13,21]. MSCs in the TME stimulate angiogenesis and promote cell growth. Furthermore, MSCs promote cancer metastasis, prevent cancer cells from undergoing apoptosis, and play a role in therapeutic resistance [22].

A previous study has shown that MSCs (of adult or fetal origin), mixed with tumor cells, promote tumor growth in vivo [23]. MSCs within tumors secrete C–C motif chemokine ligand 5 (CCL5) to stimulate breast cancer cell migration, invasion, and metastasis [24]. We have shown that ovarian MSCs secrete high levels of IL-6, promoting the proliferation, anchorage-independent growth, tumor sphere formation, and tumorigenesis of ovarian cancer cells [25]. Another study has shown that endometrial MSCs can promote endometrial cancer growth via reciprocal crosstalk between C–X–C motif chemokine 12 (CXCL12)/C–X–C chemokine receptor type 4 (CXCR4) and TGF-β1 [26]; however, the studies above did not investigate the interaction between BMSCs and ovarian cancer cells.

Understanding the mechanisms of tumor survival in the TME is essential for developing effective treatments for ovarian cancer. Toward this, investigating the interactions between BMSCs and tumor cells will be essential for designing future therapeutic strategies. Understanding these mechanisms will illuminate potential therapeutic avenues that may halt ovarian cancer progression. Here, we investigated the interactions among BMSCs and ovarian cancer cells.

## 2. Results

### 2.1. Characteristics of BMSCs Derived from Bone Marrow

The characteristics of BMSCs were evaluated through morphology, surface markers, and differentiation capabilities. The morphology of BMSCs was spindle-shaped, which was MSCs’ typical morphology (Figure 1A). The surface markers of BMSCs were negative for CD34, CD45, and HLA-DR and positive for CD29, CD44, CD73, CD90, and HLA-ABC (Figure 1B), which fulfilled the typical MSCs’ surface markers [27]. Tri-lineage differentiation (adipogenesis, osteogenesis, and chondrogenesis) was typical for MSCs [28]. After two weeks of adipogenesis, the cellular structure of BMSCs changed to a larger round shape, with neutral lipid vacuoles stained with Oil Red O (Figure 1C), as expressed by the quantification of Oil Red staining (*p* < 0.001 compared to the control, Figure 1C). After two weeks of osteogenic induction, these cells exhibited an osteoblastic morphology, characterized by mineralized matrices and being positive for Alizarin Red S staining, indicating the presence of calcium deposits in the cells (Figure 1D), as indicated by the quantification of the amount of Alizarin Red (*p* < 0.001 compared to the control, Figure 1D). After three weeks of chondrogenic differentiation, the differentiated cells showed positive Alcian blue staining (Figure 1E), as indicated by the quantification of Alcian blue (*p* < 0.001 compared to the control, Figure 1E). Overall, BMSCs exhibited typical MSC characteristics, such as spindle-shaped cell morphology, surface markers, and tri-lineage differentiation capabilities.

### 2.2. Proliferation Rates of Ovarian Cancer Cells after Culturing with BMSCs’ CM

Next, the effect of culture in BMSCs’ CM on ovarian cancer cell proliferation was evaluated. An XTT assay was used to quantify the amount of cancer cell growth on days 0, 3, 5, and 7. The CM was changed every three days. The fibroblasts were used as a control due to their similarity to MSCs (mesenchymal origin, ability to secrete cytokines and growth factors), tissue microenvironmental mimicry (a component of the TME), baseline comparison, and the interpretation of the results [29]. The CM derived from the BMSCs or FB02 significantly increased the proliferation of SKOV3 cells on days 5 and 7 compared to SKOV3 cells without co-culturing (*p* < 0.001, Figure 2A). Adding IL-6 (3 ng/mL), MCP-1 (3 ng/mL), or a combination also increased SKOV3 proliferation by more than SKOV3 alone (*p* < 0.001 on day 5, Figure 2B). In summary, the effect of BMSCs’ CM was that it could promote SKOV3 proliferation. The cytokines in BMSCs’ CM, like IL-6 and MCP-1, could also significantly promote SKOV3 proliferation.

### 2.3. Increase in Cancer Cell Migration and Invasion after Culturing with BMSCs’ CM

Transwell migration and invasion assays were performed to determine the effect of BMSC-CM on the migration and invasion of SKOV3. SKOV3 was plated on the upper well of the transwell, and the CM was placed in the bottom well. Transwell migration and invading cells were counted after the experiment was finished. Culturing with BMSCs’ CM showed significantly increasing SKOV3 migration and invasion than without co-culturing and with fibroblasts (*p* < 0.001, Figure 3). Above all, BMSCs’ CM enhanced the migration and invasion of SKOV3. The interactions and signaling between cancer cells and BMSCs indicate the influence of the TME on cancer cell behavior.

### 2.4. Increase in the Colony Formation of Cancer Cells after Culturing with BMSCs’ CM

An anchorage-independent growth assay was performed to determine the clonogenicity of SKOV3 after adding BMSC-CM. Five hundred cancer cells were plated, and BMSC-CM and FB02-CM were added to the soft agar. After 14 days of culturing, the final colony numbers were counted (Figure 4A). Culturing with BMSC-CM significantly increased the number of colonies formed by SKOV3 compared to those without co-culturing and with a FB02 CM (*p* < 0.001, Figure 4B). BMSCs have the clonogenic potential of cancer cells, highlighting the impact of the TME on cancer cell behavior and growth.

### 2.5. Increase in the Size and Proliferation Rate of SKOV3 Tumors after Co-Injection with BMSCs

Next, we checked the in vivo effects of BMSCs. The gross appearance of the xenograft tumors for SKOV3, or those co-injected with FB02 or BMSCs, is shown in Figure 5A. The growth curve of the xenografts from day 18 to day 39 is depicted in Figure 5B. It demonstrated a significant increase in tumor growth on day 39 in the SKOV3 + BMSC group mice compared to SKOV3 cells injected alone or injected with fibroblasts (*p* < 0.001, Figure 5B). The histology of xenografts from the three groups, as seen in Figure 5C, did not show differences among the three groups. Immunohistochemistry for P53 and WT1 in the tumor tissues did not differ significantly among the groups (Figure 5D,E). The protein levels of CK7, PAX8, and WT1 of xenograft tumors were also no different among the three groups (Figure 5F). There was no expression of the P53 protein in cell lines and tumor tissues (Figure 5G). These results suggested that BMSCs promoted SKOV3 tumorigenesis in vivo, as evidenced by the increased tumor size and proliferation rate.

### 2.6. BMSCs Increased the Phosphorylation of p38 MAPK and GSK-3β in the Xenografts

Other than the IL-6-STAT3 signaling pathway, p38 MAPK [30] and GSK-3β [31] may enhance the proliferation and tumor growth of ovarian cancer cells, and they could be the key driving forces in ovarian cancer development. Thus, to know the proliferation signals in the tumors, p38 MAPK and GSK-3β (Figure 6) were investigated. When co-injected with BMSCs or FB02, p38 MAPK and GSK3β phosphorylation increased. These results indicated that the presence of BMSCs or FB02 in the TME increased proliferation signals, suggesting potential signaling pathway alterations due to the presence of these cells.

### 2.7. Variation in Cytokine Expression after the Co-Culture of SKOV3 with BSMCs or FB02

Cytokines were investigated to determine the influence of BMSCs in promoting proliferation. The dot plot of the cytokine array is illustrated in Figure 7. BMSC expressed IL-6, while FB02 did not. Both BMSCs and FB02 increased the IL-6 secretion of SKOV3 after co-culturing. MCP-1 was only secreted in BMSCs and FB02 but not in SKOV3. The secretion of MCP-1 in SKOV3 significantly increased after co-culturing with BMSCs and FB02. BMSCs and FB02 did not secrete GRO, GROα, and IGFBP6. GRO and IGFBP6 in SKOV3 were increased after co-culturing (Figure 7).

IL-6 and MCP-1 had increased secretion in SKOV3 + BMSCs and SKOV3 + FB02 compared to SKOV3 alone. Quantifying the cytokine array showed increased IL-6 and MCP-1 levels in SKOV3 co-cultured with BMSCs (Figure 8A,B). The concentrations of MCP-1 and IL-6 of BMSCs were 826.5 ± 36.9 pg/mL and 888.1 ± 11.4 pg/mL, respectively. An ELISA demonstrated a significant increase in the concentration of MCP-1 (1142.2 ± 87.6 vs. 10.5 ± 1.6 pg/mL, *p* < 0.01, Figure 8C) and IL-6 (1142.2 ± 19.1 vs. 589.5 ± 1.0 pg/mL, *p* < 0.001, Figure 8D) in the CM of the co-culture with BMSCs compared to those without co-culturing. These findings indicated that the interaction between SKOV3 cells and BMSCs increased cytokine secretion, particularly for MCP-1 and IL-6, which might be responsible for the effects observed after co-culturing with SKOV3 cells.

## 3. Discussion

BMSCs were established with a characterized morphology, surface marker expression, and tri-lineage differentiation potential. Ovarian cancer cells (SKOV3) cultured with a CM from BMSCs showed increased migration, invasion, and colony formation, indicating the role of the tumor microenvironment in influencing cancer cell behavior. BMSCs promoted SKOV3 tumorigenesis in NOD-SCID mice, increasing tumor growth. The co-injection of BMSCs increased the phosphorylation of p38 MAPK and GSK-3β in SKOV3 tumors. The co-culturing of SKOV3 cells with BMSCs led to an increase in the expression of cytokines, especially MCP-1 and IL-6.

MSCs exert a range of effects on the progression and development of tumors by fostering stem-cell-like characteristics in tumor cells, facilitating their migration, promoting angiogenesis, suppressing immune responses, and inducing drug resistance [20]. In the current study, BMSCs could promote SKOV3 cell proliferation, migration, invasion, colony formation, and xenograft growth. Nevertheless, similar expressions of WT1 in the xenografts were noted in the xenografts with or without co-injection with BMSCs. WT1 are prognostic biomarkers in ovarian cancer [32]; however, their expressions were not affected by BMSCs. Subsequently, the increased phosphorylation of p38MAPK and GSK-3β was found in the xenograft co-injected with BMSCs. The activation of p38MAPK potentially influences GSK-3β and downstream β-catenin signaling pathways [33,34]. The activity of GSK-3β played a role in the proliferation of human ovarian cancer cells in laboratory settings (in vitro) and within living organisms (in vivo) [35]. Our results suggested that BMSCs enhance ovarian tumor progression through these pathways.

In addition to the above signaling pathways, cytokines derived from BMSCs may affect tumor growth [36]. Our previous reports showed that IL-6 from ovarian MSCs and CXCL12 from endometrial MSCs have tumor-promoting effects in ovarian and endometrial cancers [25,26]. This cascade of cytokine release can yield various effects, such as chemoresistance promotion, resistance to apoptosis, invasion, and angiogenesis, through the overexpression of vascular endothelial growth factor [36]. It can support the growth of metastases at distant sites [36]. Studies have demonstrated that IL-6 can activate signaling pathways, including JAK2/STAT3, promoting tumor proliferation [37]. MCP-1 could induce ERK/GSK-3β/Snail signaling to promote the epithelial–mesenchymal transition and migration of breast cancer cells [38]. MCP-1 could activate p38 MAPK signaling to stimulate MMP-9 expression [39]. In the current study, IL6 and MCP-1 levels were higher in SKOV3 cells co-cultured with BMSCs than those without co-culturing. Thus, these two cytokines may play a role in tumor progression.

MSCs co-cultured with ovarian cancer cells exhibit several significant effects that can influence tumor progression and cancer cell behavior [40]. MSCs can enhance the proliferation of ovarian cancer cells [25]. The presence of MSCs in the tumor microenvironment often leads to an increase in cancer cell division and growth, contributing to the expansion of the tumor [41,42]. MSCs are known to secrete various growth factors that promote the formation of new blood vessels (angiogenesis) [43]. This provides the tumor with a greater supply of oxygen and nutrients, facilitating further growth and survival of the cancer cells. Co-culture with MSCs can increase ovarian cancer cells’ migratory and invasive abilities [44]. This is due to the secretion of matrix metalloproteinases (MMPs) and other enzymes by MSCs, which degrade the extracellular matrix and allow cancer cells to invade surrounding tissues [45]. MSCs can induce epithelial–mesenchymal transition in ovarian cancer cells, a process by which epithelial cells acquire mesenchymal properties [46]. This transition is associated with increased mobility and invasiveness, facilitating metastasis. MSCs can secrete cytokines and growth factors that activate survival pathways in ovarian cancer cells, making them more resistant to chemotherapy [25,47,48]. This includes activating pathways such as Hedgehog, which promote cell survival and resistance to cell death [49]. MSCs secrete a range of paracrine factors that can stimulate ovarian cancer cells directly or through modification of the tumor microenvironment [50,51]. These factors include cytokines, chemokines, and growth factors [51]. In the current study, our findings align with the former group, indicating that BMSCs induce malignant behavior in ovarian cancer cells. This highlights the diverse and context-dependent interactions between MSCs and ovarian cancer cells, emphasizing the need for further investigation to elucidate the underlying mechanisms and potential therapeutic implications.

This study had some limitations, including its focus on short-term effects and the use of specific ovarian cancer cell lines; thus, we recognize the need for more clinically relevant investigations. Additionally, considering a broader range of ovarian cancer cell subtypes and evaluating the in vivo effects on poorly responding cell lines would contribute to a more comprehensive understanding of the roles played by BMSCs and ovarian cancer cells in terms of their influence on tumor behavior and the TME.

## 4. Materials and Methods

### 4.1. Isolation of Bone Marrow Mesenchymal Stem Cells

One previously derived BMSC line (BMSC-05) was used. The cells were obtained from a female patient, aged 76, who suffered from a left femoral neck fracture and received bipolar hemiarthroplasty surgery. The derivation protocol was as follows: The bone marrow was aspirated from a patient who underwent orthopedic surgery. Patient consent was obtained, and the ethical and Declaration of Helsinki guidelines were followed. The study was approved by the Research Ethics Committee of Hualien Tzu Chi Hospital (approval number: IRB111-183-C).

Five milliliters of bone marrow aspirate was mixed with 5 mL of phosphate-buffered saline (PBS; Sigma, St. Louis, MO, USA) and centrifuged at 1200× *g* for 6 min. The supernatant was then removed. The pellet was plated on a 10 cm culture dish and supplemented with a culture medium consisting of alpha-minimal essential medium (Gibco, Waltham, MA, USA), 15% fetal bovine serum (FBS, Gibco), and 1% penicillin/streptomycin (Sigma). The BMSCs were maintained in this culture medium until they reached 70–90% confluence. The cells were harvested by adding trypsin/EDTA (Sigma) and replating at a 1:3 ratio in new culture dishes.

### 4.2. Primary Human Skin Fibroblasts

Human skin fibroblasts were purchased from the Bioresource Collection and Research Center (BCRC, 08C0011, Hsinchu, Taiwan). The culture medium comprised Dulbecco’s modified Eagle’s medium (DMEM, Gibco) plus 15% FBS (Gibco) and 4 mM L-glutamine (Sigma), adjusted to contain 1.5 g/L sodium bicarbonate (ThermoFisher, Waltham, MA, USA) and 4.5 g/L glucose (Merck, Darmstadt, Germany). The culture medium was renewed after every 2–3 days. After the fibroblasts reached 70–90% confluence, they were harvested and passaged at a 1:3 ratio. Fibroblasts were used as controls in the experiment involving the interaction of BMSCs with SKOV3 tumor cells to provide a baseline comparison for assessing the specific effects of BMSCs. Fibroblasts, a common stromal cell type found in the tumor microenvironment [52], serve as a relevant control to evaluate the impact of BMSCs on tumor proliferation independent of generic stromal cell effects.

### 4.3. Culturing of Ovarian Cancer Cells

Ovarian cancer cell lines, SKOV3 cells, were purchased from the American Type Culture Collection (ATCC; Manassas, VA, USA) and used for co-culture experiments. SKOV3 is a cell line with an epithelial morphology derived from the ovary of a 64-year-old Caucasian female diagnosed with ovarian adenocarcinoma. Cancer cells derived from ascites induced by SKOV3 cells exhibit heightened malignant traits, such as accelerated growth, increased colony formation capacity, and reduced host animal survival [52]. The SKOV3 cells were cultured in Roswell Park Memorial Institute (RPMI)-1640 medium (Gibco) containing 10% FBS (Gibco) and 1% penicillin/streptomycin (Sigma) [53].

All cells were cultured in 5% CO_2_ at 37 °C in a humidified atmosphere.

### 4.4. Flow Cytometry

The BMSCs were stained for 30 min with fluorescent-dye-conjugated antibodies on ice and washed twice with PBS. After blocking with PBS containing 2% FBS, the BMSCs were analyzed using fluorescence-activated cell sorting (FACSCalibur; BD Biosciences, Franklin, NJ, USA). The following antibodies were used: anti-CD29, anti-CD34, anti-CD44, anti-CD45, anti-CD73, anti-CD105, anti-HLA-ABC, and anti-HLA-DR (all from BD Biosciences). These markers are typical surface markers for BMSCs.

### 4.5. Cell Proliferation Assay

Cell proliferation was evaluated using an XTT assay (Biological Industries Ltd., Beit Haemek, Israel) [54]. SKOV3 cells were plated at 2000 cells/cm^2^ density with or without a 100% conditioned medium (CM) of the BMSCs in 96-well plates. The CM was derived from 5 × 10^5^ BMSCs cultured for 48 h. The CM was changed every three days. The cancer cells were harvested on days 0, 3, 5, and 7. IL6 (3 ng/mL, PeproTech, Cranbury, NJ, USA), MCP-1 (3 ng/mL, PeproTech), or a combination was added to the culture medium to test their effects on the proliferation of SKOV3 cells on days 0, 1, 3, and 5. The XTT solutions and N-methyl dibenzopyrazine methyl sulfate (PMS; Biological Industries Ltd.) were thawed immediately before use in a 37 °C water bath. PMS was mixed with the XTT solution immediately before application. Subsequently, 50 μL of the XTT/PMS mixture was added to each 100 μL of culture. Following an incubation period of 2–5 h at 37 °C, the optical density (O.D.) of the wells was measured using a spectrophotometer (ELISA reader, DYNEX MRX II; Dynex Technologies, Chantilly, VA, USA) at a wavelength of 450 nm with a reference wavelength of 650 nm. All the experiments were conducted in triplicate.

### 4.6. Tri-Lineage Differentiation

Tri-lineage differentiation (adipogenesis, osteogenesis, and chondrogenesis) is a typical characteristic of MSCs [28].

#### 4.6.1. Adipogenesis

The BMSCs were seeded at a density of 5 × 10^4^ cells in 1 well of a 12-well plate with an adipogenic medium (DMEM (Gibco) supplemented with 10% FBS (Gibco), 1 μmol/L dexamethasone (Sigma), 5 μg/mL insulin (Sigma), 0.5 mmol/L isobutylmethylxanthine (Sigma), and 60 μmol/L indomethacin (Sigma). Adipogenesis was induced for 14 days, and the medium was changed every 3 days. After 14 days, the differentiated cells were stained with Oil Red (Sigma) for 15 min. The Oil Red dye was eluted from the stained cells using 100% isopropanol (Honeywell, Charlotte, NC, USA). The O.D. value of Oil Red droplets was detected at 510 nm [55] using a spectrometer (BioTek Synergy HTX, Agilent, Santa Clara, CA, USA). The experiments were repeated thrice and quantified as the mean and standard deviation.

#### 4.6.2. Osteogenesis

The BMSCs were seeded at a density of 1 × 10^5^ cells in 1 well of a 12-well plate with an osteogenic medium ((DMEM (Gibco) supplemented with 10% FBS (Gibco), 0.1 μmol/L dexamethasone (Sigma), 10 mmol/L β-glycerol phosphates (Sigma), and 50 μmol/L ascorbate (Sigma). The medium was changed every 3 days, and the cells were differentiated for 14 days. After 14 days, the differentiated osteoblasts were stained with 2% Alizarin Red (Sigma) for 15 min [56]. The Alizarin Red dye was eluted from the stained cells using 100 mM cetylpyridinium chloride (Sigma). The O.D. value of Alizarin Red was detected at 562 nm. The experiments were repeated thrice and quantified as the mean and standard deviation.

#### 4.6.3. Chondrogenesis

The BMSCs were seeded at a 2.5 × 10^5^ cells/mL density in 1 well of a 12-well plate with a chondrogenic medium. The medium consisted of DMEM (Gibco), 10% FBS (Gibco), 10 ng/mL TGF-β1 (Pepro Tech, Rocky Hill, NJ, USA), 6.25 μg/mL insulin (Sigma-Aldrich), and 50 μg/mL ascorbic acid-2-phosphate (Sigma-Aldrich). The medium was changed every 3 days. The chondrogenic differentiation took 21 days; the differentiated chondrocytes were stained with Alcian blue (Sigma) for 30 min. The Alcian blue dye was eluted from the stained cells using 6 M guanidine hydrochloride (Sigma). The stained cells were observed under a microscope (Nikon, Tokyo, Japan). The O.D. value of Alcian blue was detected at 595 nm. The experiments were repeated thrice, and the data were quantified using the mean and standard deviation.

### 4.7. Migration Assay [57]

SKOV3 cells (5 × 10^4^ cells) were seeded in the upper well of a transwell Boyden chamber (24 wells; pore size, 8 μm; Costar, Corning, Corning, NY, USA). The lower well of the transwell contained a CM of the BMSCs or fibroblasts (FB02) collected after culturing for 48 h. The cancer cells in the upper well migrated to the membrane insert after 18 h. Using a cotton-tipped applicator, non-migrated cells from the apical side of the transwell insert membrane were delicately eliminated. The membranes were stained with crystal violet (Sigma) for 20 min and counted under a bright-field microscope (Axiovert 25C, Zeiss, Gottingen, Germany). CMs from fibroblasts were used as the controls. During migration experiments, cells did not uniformly penetrate the membrane on the other side; therefore, after staining with crystal violet, the areas of the membrane containing cells were initially observed at 40× magnification. After counting, the results from another membrane were then examined. Once the cell counts for all groups were completed, statistical graphs were generated to compare the proportions of cells between different groups (positive-stained cells/view).

### 4.8. Invasion Assay [58]

SKOV3 cells (5 × 10^4^ cells) were seeded on Matrigel-coated inserts in a 24-well plate (BD Biocoat Matrigel Invasion Chamber; BD Bioscience, Bedford, MA, USA). The CM from culturing the BMSCs and FB02 was loaded into the bottom well. The invasion assay was performed for 24 h. Then, Matrigel from the interior of the transwell insert was delicately yet securely eliminated using a cotton-tipped applicator. The process was repeated with a new cotton-tipped applicator to guarantee the removal of all Matrigel and any remaining non-migratory cells. The membrane was removed, the cells were plated onto a slide, and the invaded cells were observed. The cells were counted in three fields, and data were averaged to obtain the mean cell count per field. Fibroblasts were used as the controls. The counting method was described in the above section.

### 4.9. Anchorage-Independent Growth [59]

The soft agar assay for anchorage-independent growth was performed using 500 SKOV3 cells in one well of a 24-well dish. These cells were suspended in 0.5 mL of growth medium or conditioned medium derived from BMSCs or FB02 supplemented with 0.7% agarose (Sigma) and then layered onto a 0.25 mL base of 0.8% agarose. Media was replaced two times weekly. After 14 days of incubation, the colonies were counted. Colonies larger and smaller than 50 μm were counted using bright-field microscopy. Each experiment was conducted in triplicate.

### 4.10. Animal Xenograft Experiment [25]

To know the role of BMSCs and fibroblasts in promoting tumor growth, a xenograft experiment (SKOV3 alone, SKOV3 + BMSCs, and SKOV3 + FB02) was performed.

The Animal Research and Care Committee of Hualien Tzu Chi Hospital approved the animal experiments (No. 111-31). Female six-week-old non-obese diabetic severe combined immune-deficient (NOD-SCID) mice (strain name: NOD. CB17-Prkdcscid/Jtcu) were housed in the Animal Center of the Tzu Chi University. The relative humidity was maintained at 45–65%, and cages were maintained at 20–24 °C. The mice were kept in pathogen-free rooms that adhered to the recommendations outlined in the ARRIVE guidelines. The animals were accommodated in conventional cages, provided unrestricted access to water and food, and kept under a 12 h light–dark cycle.

In total, nine mice were used. Tumor cells (SKOV3, unknown passage, 1 × 10^5^ cells) alone or in combination with BMSCs (P7, 3 × 10^5^ cells) or fibroblasts (P6, 3 × 10^5^ cells) (n = 3 each) were subcutaneously injected into the back area of the mice. The transplanted cells were cultured for seven days to proliferate to the needed cell numbers before transplantation in the respective culture medium. Cells in 100 μL of culture medium were mixed with 100 μL of Matrigel (BD Bioscience Growth Factor Reduced BD Matrigel™ Matrix, BD) [60]. Mice were anesthetized with 2% isoflurane and 2 L/min oxygen inhalation [61].

The mice were sacrificed when the tumor reached 500 mm^3^ in size [62]. The tumors were imaged, measured, and weighed. Tumor volume was recorded weekly using the following equation: volume = (width)^2^ × length/2 [63]. This experiment was conducted 39 days after cell injection.

### 4.11. Histological Examination

Xenograft tumor tissues were fixed in 4% paraformaldehyde and sectioned at a thickness of 4 μm. Hematoxylin (5 min) and eosin (30 s) (H&E, Sigma) staining was performed for histological examination. Tumor sections were imaged at 200× magnification (Nikon). Nuclear morphology, cell density, and mitotic figures were recorded.

### 4.12. Immunohistochemistry (IHC)

The paraffin-embedded tissue blocks were sectioned into 5 μm slices. For deparaffinization and rehydration, the slides were placed in a 60 °C oven for 30 min. They were then immersed in two changes of xylene for 5 min each. Subsequently, the slides underwent two changes each of fresh absolute ethanol, 90% ethanol, and 80% ethanol, all for 5 min. Following this, the slides were rinsed under gently running tap water for 30 s and placed in a TBS wash bath for further rehydration for 5 min. All steps were performed at room temperature.

Antigen retrieval was conducted by soaking the slides in sodium citrate buffer (10 mM Sodium Citrate, 0.05% Tween 20, pH 6.0) and incubating them at 120 °C in a pressure cooker for at least 15 min. Once cooled to room temperature, the slides were covered with 3% hydrogen peroxide and incubated for 10 min at room temperature. After incubation, the slides were washed with TBS for 5 min. They were then pre-incubated with 5% BSA (bovine serum albumin, Sigma) for 10 min before applying the primary antibody.

For immunohistochemistry (IHC), Anti-P53 (1:100, 2527, rabbit mAb, Cell Signaling, Danvers, MA, USA), and WT1 (1:100, BSB 6033, Bio SB), monoclonal antibodies were used. The slides were incubated with primary antibodies overnight at 4 °C, followed by three washes with TBS, each for 5 min. Next, horseradish peroxidase (HRP) polymer-conjugated anti-rabbit or anti-mouse secondary antibody (Thermo Fisher Scientific) was applied for 1 h at room temperature, followed by three washes with TBS, each for 5 min. Diaminobenzidine tetrahydrochloride (DAB) Quanto Chromogen and DAB Quanto substrate mix (Thermo Fisher Scientific) were applied to the slides for 10 min at room temperature, followed by three washes with distilled water, each for 5 min. Hematoxylin was then applied for 2 min to stain the nucleus, and the slides were rinsed with distilled water. After dehydration (by immersing the slides sequentially into 80%, 90%, and absolute ethanol and xylene, reversing the rehydration steps) and mounting, the stained sections were photographed using a light microscope (Nikon TE2000-U fitted with a digital camera [Nikon DXM1200F], Nikon, Tokyo, Japan). Positive-stained cells (brown color in the nucleus) were counted in 100 cells from five randomly selected fields.

### 4.13. Cytokine Array

The medium derived from 1 × 10^6^ BMSCs or fibroblasts cultured in a 10 cm dish for two days was collected as the control. The medium derived from BMSCs or fibroblasts co-cultured with SKOV3 cells for two days was used as the experimental group. The medium was passed through a 0.22 μm cell strainer to remove the cell debris. Cytokine levels in the medium were measured using a cytokine array kit (ab193656; Abcam, Cambridge, UK), which contained 120 types of cytokines. The cytokines were quantified by spot densitometry with ImageJ software (version 1.54i, NIH, Bethesda, MA, USA).

### 4.14. Enzyme-Linked Immunosorbent Assay (ELISA)

The concentrations of cytokines in the medium were quantified using ELISA kits for IL-6 (#BMS213-2, Invitrogen, Carlsbad, CA, USA) and MCP-1 (#BMS281, Invitrogen). Corning transwell plates (pore size 8.0 μm; Corning, NY, USA) were used for the co-culture experiments. The upper chamber contained 5 × 10^4^ BMSCs or FB02 in 0.3 mL of a culture medium, while the bottom chamber contained 1 × 10^5^ SKOV3 cells in 0.6 mL of a culture medium and co-cultured for 24 hrs. Fibroblasts were used as the controls. The medium derived from BMSCs or BMSCs co-cultured with SKOV3 was incubated on an antibody-coated plate for one hour. After washing, the plates were incubated with biotin-conjugated secondary antibody for 2 h, streptavidin–horseradish peroxidase (HRP, Abcam) for 30 min, and chromogen for 10 min. After the reaction was stopped, the plates were analyzed using an ELISA reader at 450 nm as the primary wavelength and 620 nm as the reference wavelength. The difference between OD450 and OD620 and the blank were fitted into the standard curve to determine the target protein concentration. The value was normalized with cell numbers.

### 4.15. Western Blotting

Western blotting was used to evaluate the proliferation signaling pathway. The expression of p38MAPK and GSK-3β in the xenograft was checked with or without co-injected BMSCs or FB02. Tissues were lysed using a protein lysis buffer (Sigma-Aldrich) to extract the proteins. The total protein content was determined using a Bradford assay before analyzing the tumor lysates using SDS-PAGE and Western blotting. A protein-binding dye (Coomassie Brilliant Blue G-250, Sigma) bonded to proteins for 10 min in the sample, resulting in a color change. The intensity of the color change was proportional to the protein concentration, which was measured spectrophotometrically at a wavelength of around 595 nm. The proteins were separated using 10% sodium dodecyl sulfate-polyacrylamide gel electrophoresis. Briefly, 80 μg protein was aliquoted to 4 SDS-PAGE, and thus, 20 μg of each sample was loaded for the experiment. After transferring the proteins to a polyvinylidene fluoride (PVDF) membrane (Bio-Rad, Hercules, CA, USA), the first membrane was used for detecting p-GSK3β, then stripping the membrane, and further detecting GSK-3β; the secondary membrane was used for detecting p-p38, then stripping the membrane, and further detecting p38; the third membrane was used for detecting p53, then stripping the membrane, and further detecting WT1; and the final membrane was used for detecting Actin. The primary antibodies (against p38MAPK (8690, Cell Signaling), GSK-3β (9315, Cell Signaling), p-p38MAPK (4511, Cell Signaling), p-GSK-3β (9323, Cell Signaling), CK7 (A2574, Abclonal, Woburn, MA, USA), PAX8 (A1009, Abclonal), WT1 (GTX131203, GeneTex), GAPDH (2118, Cell Signaling), and actin (4967, Cell Signaling), p53 [2527, rabbit mAb, Cell Signaling]) were diluted 1:2000 and then incubated with the membranes overnight at 4 °C. Subsequently, the membranes were exposed to 1:5000 dilution of a secondary antibody (anti-rabbit or anti-mouse immunoglobulin G-HRP; Amersham GE, Taipei, Taiwan). The HRP signals of p-GSK-3β were detected using SuperSignal West Femto Maximum Sensitivity Substrate electrochemiluminescence (ECL) kit (Thermo Fisher Scientific), and the others were detected using SuperSignal West Pico PLUS Chemiluminescence Substrate kit (Thermo Fisher Scientific). The protein bands underwent scanning by iBright image system (Thermo Fisher Scientific), and their intensities were quantified utilizing ImageJ software (NIH). The results of quantification of total GSK-3β and total p38 were first normalized with Actin (nor-Actin), and each sample was further compared with SKOV3 and its ratio to SKOV3 calculated. The results of quantification of phospho-protein were normalized with its nor-Actin of total form, and each sample was further compared with SKOV3 and the ratio to SKOV3 calculated.

### 4.16. Statistical Analysis

Student’s *t*-tests were used to compare continuous variables between groups. Data are presented as means, along with standard deviations. The chi-square test was used to analyze categorical variables. Analysis of variance (ANOVA) with a post hoc Bonferroni test was used to analyze data derived from more than two groups. Statistical analyses were performed using SPSS software (version 24, IBM, New York, NY, USA). Statistical significance was set at *p* < 0.05.

## 5. Conclusions

Our study provides valuable insights into the complex interplay between BMSCs and ovarian cancer cells, highlighting their influence on tumor behavior and the TME. Understanding the mechanisms underlying the BMSC-mediated promotion of ovarian cancer progression opens up avenues for developing novel therapeutic strategies. Targeting specific cytokines or signaling pathways identified in our study, such as MCP-1, IL-6, p38MAPK, and GSK-3β, could potentially disrupt the tumor-promoting effects of BMSCs and inhibit ovarian cancer progression. The identified cytokines and signaling pathways could be biomarkers for assessing ovarian cancer progression and predicting patient outcomes. Monitoring the expression levels of MCP-1 and IL-6, or the activation status of p38MAPK and GSK-3β, in tumor tissues or patient serum may provide valuable diagnostic information and aid in treatment decision-making. Further research is warranted to elucidate the mechanisms underlying the crosstalk between BMSCs and ovarian cancer cells. Investigating the additional cytokines, signaling pathways, and molecular interactions involved in this interplay could uncover new therapeutic targets and refine existing treatment strategies for ovarian cancer.

## Figures and Tables

**Figure 1 ijms-25-06746-f001:**
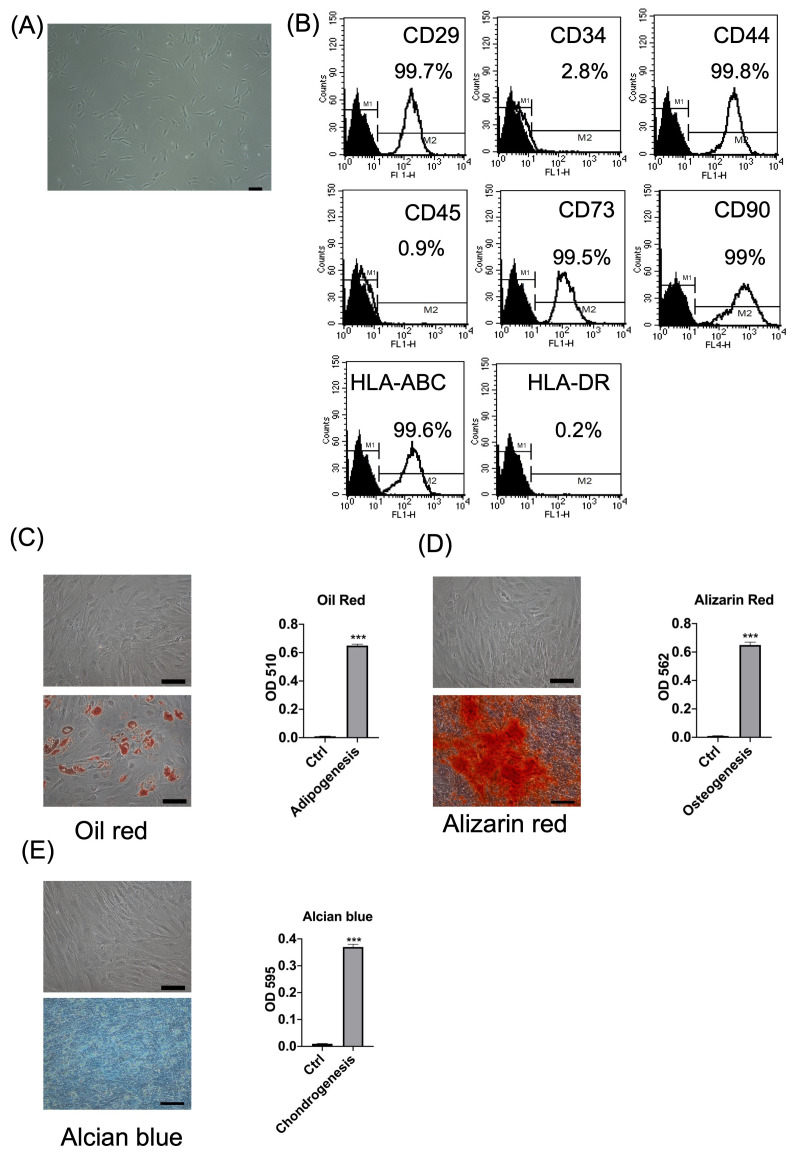
Characteristics of bone marrow mesenchymal stem cells (BMSCs). (**A**) The BMSCs showed a fibroblastic morphology. Scale bar = 1000 μm. (**B**) Cell surface markers of BMSCs. Positivity for CD29, CD44, CD73, CD90, and HLA-ABC, and negativity for CD34, CD45, and HLA-DR, were noted. (**C**–**E**) Tri-lineage differentiation of the BMSCs. (**C**) Adipogenesis was assessed using Oil Red staining and quantification. (**D**) Osteogenesis was assessed using Alizarin Red staining and quantification. (**E**) Chondrogenesis was assessed using Alcian blue staining and quantification. The upper panel of (**C**–**E**) is a negative control. *** *p* < 0.001 compared to the control. Scale bar = 100 μm in (**C**–**E**). N = 3 each in (**C**–**E**).

**Figure 2 ijms-25-06746-f002:**
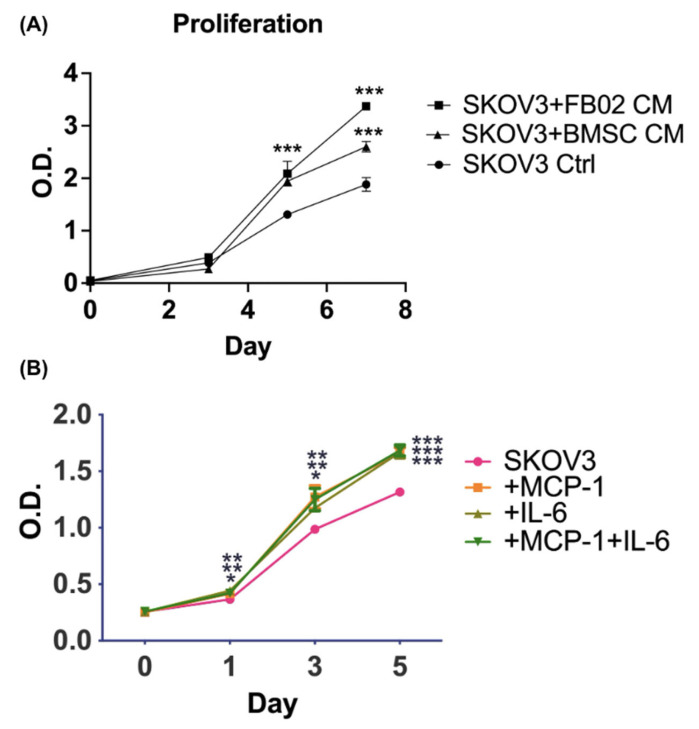
Proliferation curve. (**A**) Proliferation curve of SKOV3 cells with or without co-culturing with conditioned medium from bone marrow stem cells (BMSCs) or fibroblasts (FB02) from day 0 to day 7. *** *p* < 0.001 compared with the control. (**B**) Proliferation curve of SKOV3 with or without IL-6 (3 ng/mL), MCP-1 (3 ng/mL), or a combination (3 ng/mL each) from day 0 to day 5. *** *p* < 0.001, ** *p* < 0.01, and * *p* < 0.05. O.D.: optical density. N = 3 each.

**Figure 3 ijms-25-06746-f003:**
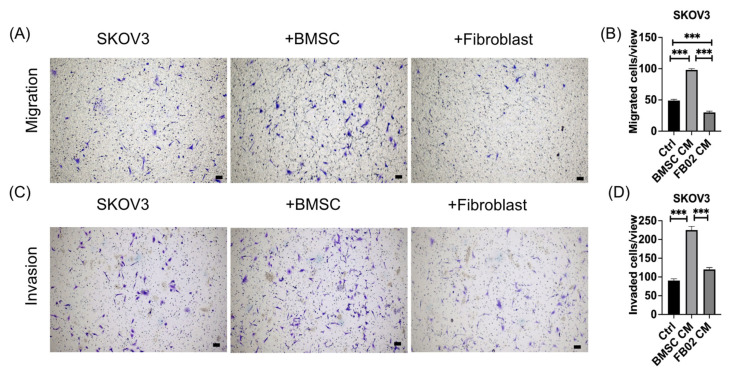
Migration and invasion assay of SKOV3 with or without the CM obtained from culturing bone marrow mesenchymal stem cells (BMSCs) or fibroblasts (FB02). (**A**) Migration assay of SKOV3. Migrated cells were stained with crystal violet. (**B**) Quantification of the number of migrating cells. (**C**) Invasion assay of SKOV3. (**D**) Quantification of the number of invading cells. Invaded cells were stained with crystal violet. *** *p* < 0.001. Scale bar = 100 μm. N = 3 each.

**Figure 4 ijms-25-06746-f004:**
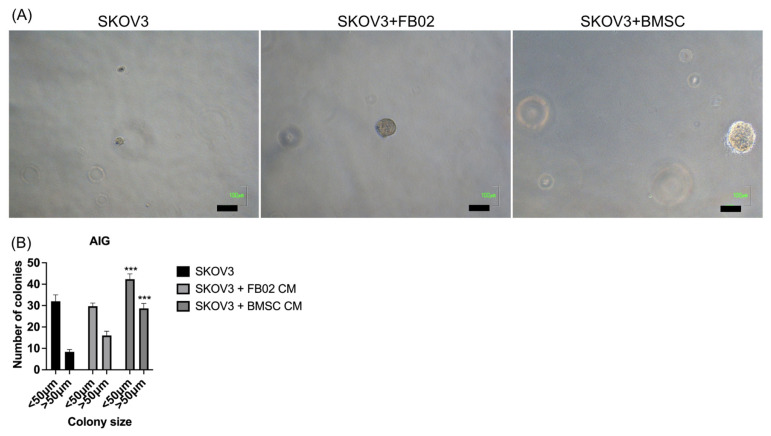
SKOV3 cells showed anchorage-independent growth. (**A**) Soft agar assay for colony formation of SKVO3 alone, SKOV3 + FB02 condition medium (CM), and SKOV3 + BMSC CM for 14 days resulted in colonies being photographed. Scale bar = 100 μm. (**B**) Numbers of colonies formed in soft agar in different groups. Bars represent means ± standard deviation. *** *p* < 0.001 compared to SKOV3 alone and SKOV3 + FB02 CM. N = 3 each.

**Figure 5 ijms-25-06746-f005:**
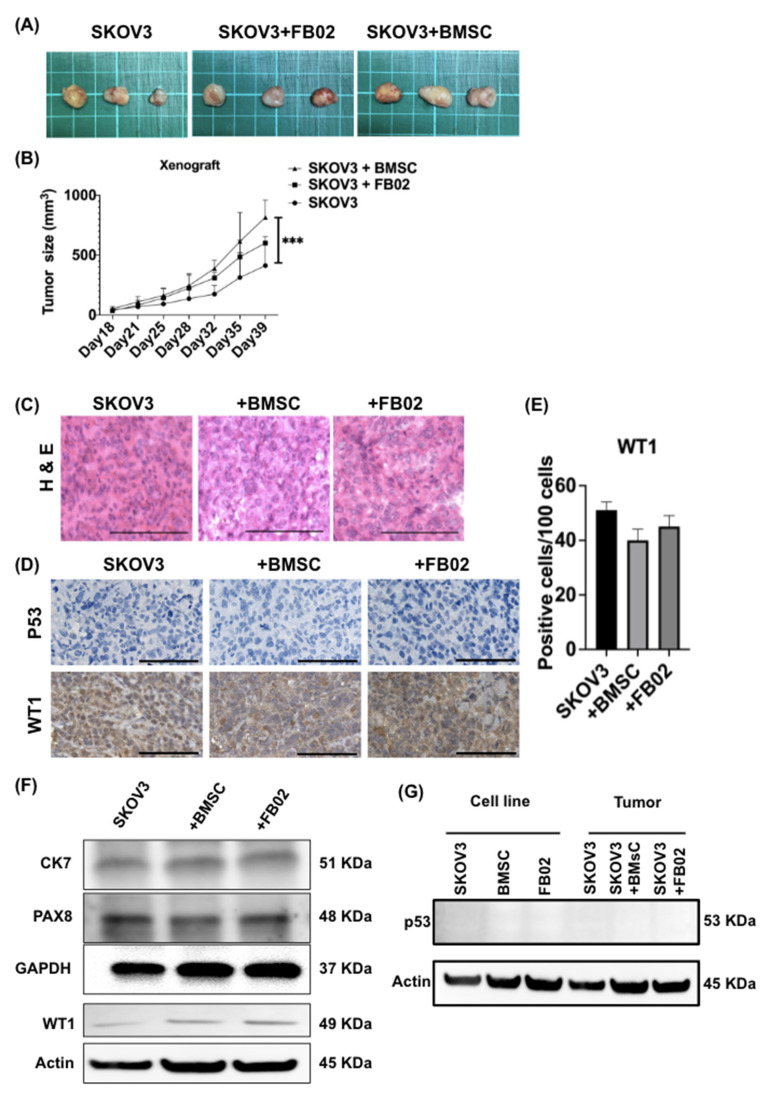
Bone marrow stem cells (BMSCs) promoted ovarian cancer cell (SKOV3) tumorigenesis in vivo. NOD-SCID female mice (5–6 weeks old) were subcutaneously injected with SKOV3 (1 × 10^5^ cells) with or without FB02 (3 × 10^5^ cells) or BMSCs (3 × 10^5^ cells) (n = 3 in each group). (**A**) Gross pictures of the xenograft tumors formed by SKOV3 without or with FB02 or BMSCs (n = 3 in each group). (**B**) The growth curve of xenograft tumors was measured after every three or four days (n = 3 in each group). *** *p* < 0.001 in the SKOV3 + BMSC group compared with that in the SKOV3 group. (**C**) H&E staining of xenograft tumors in the three groups. Scale bar = 100 μm. (**D**) Immunohistochemistry of P53 (upper panel) and WT1 (lower panel) of the tumor tissues. Scale bar = 100 μm. (**E**) Quantification of WT1 staining. No significant difference was found among the groups. The positive-stained cells (brown color in the nucleus) of 100 cells in the five randomly selected fields were calculated. (**F**) Western blot analysis for CK7, PAX8, and WT1 in the xenograft tumors of the three groups. GAPDH was used as an internal control. (**G**) Western blot of p53 protein from cell lines and tumor tissues. No expression in all cell lines and tissues was noted.

**Figure 6 ijms-25-06746-f006:**
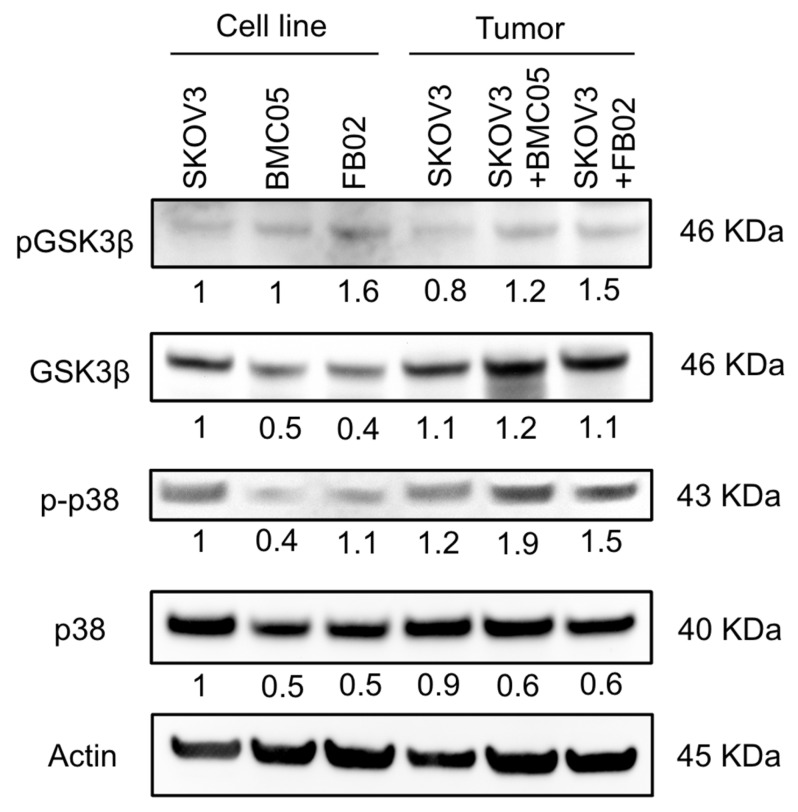
Protein expression of SKOV3 cell and xenograft tumors with or without co-injected BMSCs or FB02. p38 MAPK and GSK-3β were evaluated. Actin was used as the internal control. The number below each lane was a relative quantification of the protein content of SKOV3 cells, which was adjusted with actin and respective total protein.

**Figure 7 ijms-25-06746-f007:**
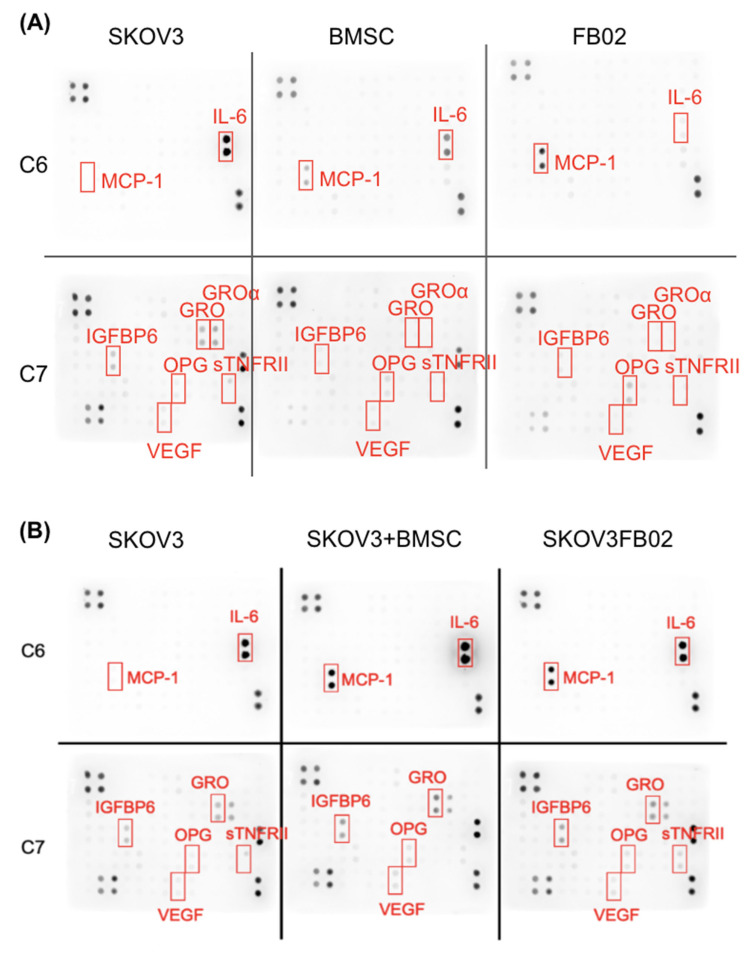
Cytokine array of conditioned medium from SKOV3, BMSC, FB02, or co-culture conditions. (**A**) Cytokine array of SKOV3, BMSC, and FB02, respectively. (**B**) Cytokine array of SKOV3, SKOV3 + BMSC, SKOV3 + FB02. MCP-1: Monocyte chemoattractant protein-1, IL-6: Interleuking-6. IGFBP6: Insulin-like growth factors binding protein-6, GRO-alpha: CXCL1, Chemokine (C–X–C motif) ligand 1, VEGF: Vascular endothelial growth factor. OPG: Osteoprotegerin. sTNFRII: soluble Tumor Necrosis Factor receptor II. C6 and C7: different cytokine combinations.

**Figure 8 ijms-25-06746-f008:**
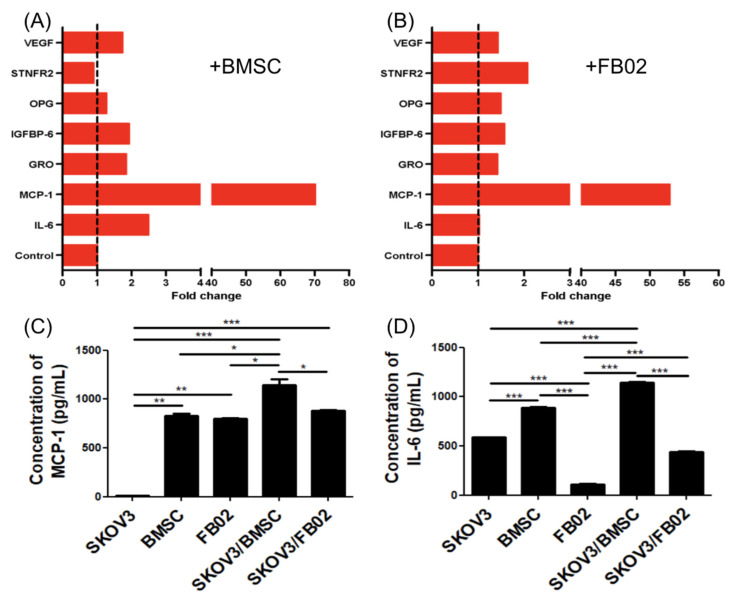
Quantification of cytokine array and ELISA of conditioned medium from SKOV3 co-cultured with BMSCs or FB02. (**A**,**B**) The cytokines were quantified by spot densitometry. Quantifying the SKOV3 + BMSCs (**A**) or FB02 (**B**) cytokine array. An ELISA of MCP-1 (**C**) and IL-6 (**D**) in a conditioned medium of SKOV3, BMSCs, FB02, SKOV3 + BMSCs, and SKOV3 + FB02. * *p* < 0.05, ** *p* < 0.01, and *** *p* < 0.001. N = 3 each in (**B**–**D**). BMSCs: bone marrow mesenchymal stem cells. FB02: fibroblasts.

## Data Availability

The data presented in this article cannot be shared because of data protection regulations. According to the Ethics Committee, only the evaluation of anonymized data was permitted for this study.

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
