# Peer review of "Bone Marrow Mesenchymal Stem Cells Promote Ovarian Cancer Cell Proliferation via Cytokine Interactions"

_ijms, 2024, doi:10.3390/ijms25126746_

Round 1
Reviewer 1 Report
Comments and Suggestions for Authors
The article by Wang K-H et al. describes in vitroand in vivowork on the impact of mesenchymal stem cells derived factors on ovarian cancer cells phenotype.
The topic is of great interest as it relates to tumor microenvironment cross-talk between stromal cell types and cancer cells, which has impact on tumor growth as well as response to treatment (Mol Cancer Res (2012) 10 (10): 1254–1264.).
Although the article addresses an important tumor biology process, it shows several technical as well as experimental flaws. Also, the English writing is sometimes not clear or the sentence meaning is different between figure legends, methods and results.
Specifically, some of the important errors identified are:
Introduction
The IJMS has a broad audience; therefore, it should be described the current knowledge on how mesenchymal stem cells are homing into tumor tissues and what roles do they play inside the tumor microenvironment with emphasis on ovarian cancer.
Materials and methods
4.3 – why are SKOV3 cells grown in RPMI while ATCC recommends McCoy’s 5a media for this cell line?
4.7.2 – The standard procedure of osteogenic differentiation in vitroconsists of seeding 5000 cells/cm2 which generate mineralized mass in 21-28 days upon induction if using bone marrow-derived adult human MSCs. Authors use ~2600 cells/cm2 and show they have obtained mineralized nodules at day 14. It contradicts the previous assessment at 21 days made in their previous paper (cited ref 11). Are they using cells from children or younger donors? They do not show any controls on cells seeded in the same density and grown in regular media.
4.7. – It is not clear how the spectrometric quantifications are performed. Are the cell monolayers extracted? Is there any standard curve used?
4.8 – It is not clear if the authors removed non-migrating cells from the upper part of the membrane before examining it under the microscope.
The colony formation assay is missing, as well as the IHC staining.
4.12 – How are the cytokines quantified? Do the authors perform spot densitometry?
4.13 – The “SKOV3 only” control is missing for ELISA determinations
4.14 – Was the total protein content determined before analyzing the tumor lysates by SDS-PAGE and Western blot?
Results
Figure 1C-E – Negative controls are lacking to show color specificity upon in vitrodifferentiation. It is not common to obtain mineralized osteoblast monolayers after only 14 days of osteoinduction.
Figure 2 – The graph is very odd – depincting “percent of day 0” versus time instead of “Number of cells counted” versus time, characteristic to cell doubling curves. One cannot assess the real impact on cell growth of the presence of conditioned media from MSC or fibroblasts.
Figure 3 – The graphs should state “migrated cells/view” and “invaded cells/view”; when counting the cells in the presented field of view, one can see that they do not match the graph representations. How were the cells counted? For example, there are ~60 cells in image C, condition +BMSC; the graph shows an average of over 200 for this condition.
Authors state unclearly and incorrectly in the results that they co-cultured SKOV3 cells with BMSC or fibroblast. They did this only for the cytokine detection assay. For all other experiments, they used conditioned media from BMSC/fibroblasts to study its effect on migration, invasion (by chemoattraction), and colony forming units capacity. This should be corrected in text.
Figure 4 – contains the repeated mistake of co-culturing BMSC and SKOV3 when they in fact measured SKOV3 colony formation upon treatement with conditioned media. The graph presents “precent” instead of “numer of colonies formed”. Method was not described.
2.5. Line 127 - Sentences are not written clearly: “compared to without coinjected or with fibroblast” instead of “as compared to SKOV3 cells injected alone or injected together with fibroblasts”
Figure 5 - It is hard to assess the contribution of BMSC or FB02 on tumor growth because a control of BMSC only or FB only cells is missing. The numbers in figure legends and those in Methods are not identical. How many cells were injected? The IHC cannot be clearly interpreted? How do the authors perform quantification? The whole method is missing from Methods. P53 and WT1 staining is done in parallel on tumors with or without co-injected stromal cells. How do authors discriminate the expression by cancer cells? What was the hypothesis and the reason of choosing these markers? A number of 3 animals per group is fairly small to observe potential significant differences.
Western blot bands from the original films were trimmed without explaining the existence of extra bands with enhanced intensity. Is the molecular weight marker analyzed correctly to determine mass?
2.6. Authors state that there is increased phosphorylated protein in the tumor lysate. Total protein is also enhanced and it seems that there was no normalization analysis performed. Are equal amounts of total protein analyzed?
Usually p-protein signal is normalized to the actin detected on the same membrane; also for total protein; then, the numbers obtained are used to determined phosphorylated/total protein ratio. Here most probably p-p38 increased when SKOV3 were co-injected with MSC as compared to SKOV3 alone; however, we cannot draw the same conclusion for pGSK-3b as the total protein is also increased…
How do they interpret the difference taking into account that the tumor contains not only cancer cells but also other type of cells that express these proteins?
2.7. Co-culture should be compared to single culture and cytokine secretion normalized first to number of cells (or total protein content of the media) at the endpoint for each sample.
Discussion
The conclusion is not correct:
“Ovarian cancer cells (SKOV3) co-cultured with BMSCs showed increased migration, invasion, and colony formation” All these experiments were performed using conditioned media and not co-cultures.
The article should be carefully revised and rewritten with careful consideration for data handling.
Given the above observations, I propose the article for major revision.
Comments on the Quality of English Language
The article should be carefully rewritten for clarity, especially while explaining comparison between different experimental conditions that are not named clearly.
Author Response
Reviewer 1
The article by Wang K-H et al. describes in vitro and in vivo work on the impact of mesenchymal stem cells derived factors on ovarian cancer cells phenotype.
The topic is of great interest as it relates to tumor microenvironment cross-talk between stromal cell types and cancer cells, which has impact on tumor growth as well as response to treatment (Mol Cancer Res (2012) 10 (10): 1254–1264.).
Response: We thank the reviewer’s comment. We have added this reference to the introduction section [ref 13].
[13] Musrap, N.; Diamandis, E.P. Revisiting the Complexity of the Ovarian Cancer Microenvironment--Clinical Implications for Treatment Strategies. Mol. Cancer Res. 2012, 10, 1254–1264.
Although the article addresses an important tumor biology process, it shows several technical as well as experimental flaws. Also, the English writing is sometimes not clear or the sentence meaning is different between figure legends, methods and results.
Specifically, some of the important errors identified are:
Introduction
The IJMS has a broad audience; therefore, it should be described the current knowledge on how mesenchymal stem cells are homing into tumor tissues and what roles do they play inside the tumor microenvironment with emphasis on ovarian cancer.
Response: We thank the reviewer’s comment. We have added a paragraph describing how MSC is homing into tumor tissues and the interaction between MSC and the tumor microenvironment. The statements reas as”Within the primary ovarian tumor microenvironment, BMSCs may migrate to the tumor stroma in response to chemotactic signals released by cancer cells or other stromal cells [11]. Once in the tumor vicinity, BMSCs can directly interact with ovarian cancer cells through cell-to-cell contact or paracrine signaling [12,13]. As ovarian cancer progresses, metastases often occur in the peritoneal cavity, including the omentum, peritoneum, and abdominal organs [14]. BMSCs may home in on these metastatic sites and contribute to the formation of a supportive microenvironment for cancer cell survival and growth [15]. The bone marrow serves as a reservoir for stem and progenitor cells, including BMSCs. Ovarian cancer cells may secrete factors that recruit BMSCs to the bone marrow niche, where they could potentially influence cancer cell behavior and contribute to the establishment of bone metastases [16]. The interaction between BMSCs and ovarian cancer cells can occur through various mechanisms, including paracrine signaling, exosome-mediated communication, and direct cell-to-cell contact [17–19].” (page 2, lines 62-74)
Materials and methods
4.3 – why are SKOV3 cells grown in RPMI while ATCC recommends McCoy’s 5a media for this cell line?
Response: We thank the reviewer’s comment. In our lab, we used RPMI to cultivate SKOV3 cells. We have added a reference that SKOV3 cells could be cultured using RPMI [ref 30]. (page 3, line 143)
[30] Chae, J.; Kim, J.S.; Choi, S.T.; Lee, S.G.; Ojulari, O.V.; Kang, Y.J.; Kwon, T.K.; Nam, J.-O. Farrerol Induces Cancer Cell Death via ERK Activation in SKOV3 Cells and Attenuates TNF-α-Mediated Lipolysis. Int. J. Mol. Sci. 2021, 22, 9400.
4.7.2 – The standard procedure of osteogenic differentiation in vitro consists of seeding 5000 cells/cm2 which generate mineralized mass in 21-28 days upon induction if using bone marrow-derived adult human MSCs. Authors use ~2600 cells/cm2 and show they have obtained mineralized nodules at day 14. It contradicts the previous assessment at 21 days made in their previous paper (cited ref 11). Are they using cells from children or younger donors? They do not show any controls on cells seeded in the same density and grown in regular media.
Response: We thank the reviewer’s comment. We are sorry the mistake was made. The cell number was 1 x 10^5. We made changes to the manuscript (page 4, line 191). We have added the picture of control (cell growth in a regular medium) (Figure 1).
Authors use ~2600 cells/cm2 and show they have obtained mineralized nodules at day 14. It contradicts the previous assessment at 21 days made in their previous paper (cited ref 11): We have added a reference to demonstration osteogenesis can be finished in 14 days (ref. 34).
[34] Sadraei, F.; Ghollasi, M.; Khakpai, F.; Halabian, R.; Jalali Tehrani, H. Osteogenic Differentiation of Pre-Conditioned Bone Marrow Mesenchymal Stem Cells with Nisin on Modified Poly-L-Lactic-Acid Nanofibers. Regen Ther 2022, 21, 263–270.
Are they using cells from children or younger donors? No. The background the BMSC donor was described in the method section 2.1. The statement reads as”The cells were obtained from a female patient, aged 76, who suffered from a left femoral neck fracture and received bipolar hemiarthroplasty surgery.” (page 3, line 105-107)
4.7. – It is not clear how the spectrometric quantifications are performed. Are the cell monolayers extracted? Is there any standard curve used?
Response: We thank the reviewer’s comment. We have added a description of how to extract the dye from staining. The statement read as” The Oil Red dye was eluted from the stained cells using 100% isopropanol”. (page 4, line 184-185)
4.8 – It is not clear if the authors removed non-migrating cells from the upper part of the membrane before examining it under the microscope.
Response: We thank the reviewer’s comment. We have added the description regarding removing non-migrating cells from the upper part of the membrane before examining it (sections 2.7 and 2.8). The statements read as”Using a cotton-tipped applicator, delicately eliminate any non-migrated cells from the apical side of the transwell insert membrane.” (page 5, line 217-219, 235-238)
The colony formation assay is missing, as well as the IHC staining.
Response: We thank the reviewer’s comment. We have added the colony formation assay and IHC staining protocol in sections 2.9 and 2.12. The statements read as”
2.9. Colony-forming assay [37]
Fifty cells were plated in a well of 6-well plate. BMSC-CM and FB02-CM collected from culturing BMSC and FB02 (from 5 × 105 cells for 48 h) were added. The medium was changed every 3 days. After 14 days of culture, the final colony-occupied area was counted. SKOV3 alone colony-occupied area was used as control. The colonies are fixed by treatment with glutaraldehyde (6.0% v/v, Sigma), followed by staining with crystal violet solution (0.5% w/v, Sigma)). Subsequently, the colony's cover area is calculated using a microscope (Nikon) and ImageJ software (National Institutes of Health, Bethesda MD, USA) [38]. SKOV3 cells were used as control (100%). (page 5, line 243-251)
2.12. Immunohistochemistry (IHC)
Anti-P53 and WT1 monoclonal antibodies (1: 100, GeneTex, Irvine, CA, USA) were used for IHC. A diaminobenzidine tetrahydrochloride substrate was used after incubation with horseradish peroxidase (HRP)-linked secondary antibody to detect reactivity. Photographs of the stained sections were recorded by a light microscope (Nikon TE2000-U fitted with a digital camera [Nikon DXM1200F], Nikon, Tokyo, Japan). The positive staining cells (brown color in the nucleus) in 100 cells in the five randomly selected fields were calculated.” (page 6, line 286-294)
4.12 – How are the cytokines quantified? Do the authors perform spot densitometry?
Response: We thank the reviewer’s comment. The cytokines were quantified by spot densitometry. (page 7, line 302-303, page 15, line 505)
4.13 – The “SKOV3 only” control is missing for ELISA determinations
Response: We thank the reviewer’s comment. We have added the SKOV3-only control in Figure 7C-D.
4.14 – Was the total protein content determined before analyzing the tumor lysates by SDS-PAGE and Western blot?
Response: We thank the reviewer’s comment. We have added the protein content assay in section 2.14. The statements read as”The total protein content was determined using a Bradford assay before analyzing the tumor lysates using SDS-PAGE and Western blotting. A protein-binding dye (Coomassie Brilliant Blue G-250, Sigma) bonded to proteins for 10 mins in the sample, resulting in a color change. The intensity of the color change was proportional to the protein concentration, which was measured spectrophotometrically at a wavelength of around 595 nm.” (page 7, lines 324-329).
Results
Figure 1C-E – Negative controls are lacking to show color specificity upon in vitrodifferentiation. It is not common to obtain mineralized osteoblast monolayers after only 14 days of osteoinduction.
Response: We thank the reviewer’s comment. We have added the picture of the control experiment (Figure 1C-E)(page 9). We also added a reference to address the 14-day osteoblast differentiation protocol [ref 34].
[34] Sadraei, F.; Ghollasi, M.; Khakpai, F.; Halabian, R.; Jalali Tehrani, H. Osteogenic Differentiation of Pre-Conditioned Bone Marrow Mesenchymal Stem Cells with Nisin on Modified Poly-L-Lactic-Acid Nanofibers. Regen Ther 2022, 21, 263–270.
Figure 2 – The graph is very odd – depincting “percent of day 0” versus time instead of “Number of cells counted” versus time, characteristic to cell doubling curves. One cannot assess the real impact on cell growth of the presence of conditioned media from MSC or fibroblasts.
Response: We thank the reviewer’s comment. We have changed the Y axis from percent to OD value (optical density from XTT assay)(Figure 2).
Figure 3 – The graphs should state “migrated cells/view” and “invaded cells/view”; when counting the cells in the presented field of view, one can see that they do not match the graph representations. How were the cells counted? For example, there are ~60 cells in image C, condition +BMSC; the graph shows an average of over 200 for this condition.
Response: We thank the reviewer’s comment. We changed graphs text to “migrated cells/view” and “invaded cells/view”. We have added the detailed counting method in the method section 2.7 and 2.8. The statements read as”During migration experiments, cells did not uniformly penetrate the membrane on the other side; therefore, after staining with crystal violet, the areas of the membrane containing cells were initially observed at 40x magnification. After counting, the results from another membrane were then examined. Once the cell counts for all groups were completed, statistical graphs were generated to compare the proportions of cells between different groups (positive-stained cells/view). Subsequently, images required for the report were captured at 100x magnification. The reason for not using 40x images was that cells were too small to be visible at this magnification.” (page 5, line 221-229)
Authors state unclearly and incorrectly in the results that they co-cultured SKOV3 cells with BMSC or fibroblast. They did this only for the cytokine detection assay. For all other experiments, they used conditioned media from BMSC/fibroblasts to study its effect on migration, invasion (by chemoattraction), and colony forming units capacity. This should be corrected in text.
Response: We thank the reviewer’s comment. We have changed the experiment description to culture with a conditioned medium derived from BMSC or FB02 (sections 3.2-3.4).
Figure 4 – contains the repeated mistake of co-culturing BMSC and SKOV3 when they in fact measured SKOV3 colony formation upon treatement with conditioned media. The graph presents “precent” instead of “numer of colonies formed”. Method was not described.
Response: We thank the reviewer’s comment. We have added the description regarding colony forming assay and calculation method in section 2.9. The statements read as”
2.9. Colony-Forming Assay [37]
Fifty cells were plated in a well of a 6-well plate. BMSC-CM and FB02-CM, collected from culturing BMSCs and FB02 (from 5 × 10^5 cells for 48 h), were added. The medium was changed every 3 days. After 14 days of culturing, the final colony-occupied area was counted. The SKOV3-alone colony-occupied area was used as the control. The colonies were fixed via fixed with glutaraldehyde (6.0% v/v, Sigma), followed by staining with crystal violet solution (0.5% w/v, Sigma) for 5 mins. Subsequently, the colony’s cover area is calculated using a microscope (Nikon) and ImageJ software (National Institutes of Health, Bethesda MD, USA) [38]. SKOV3 cells were used as the control (100%).”
2.5. Line 127 - Sentences are not written clearly: “compared to without coinjected or with fibroblast” instead of “as compared to SKOV3 cells injected alone or injected together with fibroblasts”
Response: We thank the reviewer’s comment. We have revised it accordingly. The statement reads as” It demonstrated a significant increase in tumor growth on day 39 in the SKOV3 + BMSCs group mice compared to SKOV3 cells injected alone or injected together with fibroblasts (p<0.001, Figure 5b).” (page 12, line 451)
Figure 5 - It is hard to assess the contribution of BMSC or FB02 on tumor growth because a control of BMSC only or FB only cells is missing. The numbers in figure legends and those in Methods are not identical. How many cells were injected? The IHC cannot be clearly interpreted? How do the authors perform quantification? The whole method is missing from Methods. P53 and WT1 staining is done in parallel on tumors with or without co-injected stromal cells. How do authors discriminate the expression by cancer cells? What was the hypothesis and the reason of choosing these markers? A number of 3 animals per group is fairly small to observe potential significant differences.
Response: We thank the reviewer’s comment. We are sorry for the mistake. The injected cell numbers were SKOV3 (1 × 10^5 cells) with or without FB02 (3 × 10^5 cells) or BMSCs (3 × 10^5 cells). The IHC experiment was stated in the Method section 2.12. We calculated the positive staining cells (brown color in the nucleus) in 100 cells in the five randomly selected fields (in section 2.12 and figure legend, page 13, line 463-464).
The statements read as”
2.12. Immunohistochemistry (IHC)
Anti-P53 (1:100, BSB 5845, Bio SB, Santa Barbara, CA, USA) and WT1 (1:100, BSB 6033, Bio SB) monoclonal antibodies were used for IHC. A diaminobenzidine tetrahydrochloride substrate was used after incubation with horseradish peroxidase (HRP)-linked secondary antibodies (TL-060-QHD, Thermo Fisher Scientific, Cheshire, UK) to detect reactivity. Photographs of the stained sections were recorded using a light microscope (Nikon TE2000-U fitted with a digital camera [Nikon DXM1200F], Nikon, Tokyo, Japan). The positive-stained cells (brown color in the nucleus) of 100 cells in the 5 randomly selected fields were calculated.” (page 6, line 286-294)
Western blot bands from the original films were trimmed without explaining the existence of extra bands with enhanced intensity. Is the molecular weight marker analyzed correctly to determine mass?
Response: We thank the reviewer’s comment. We have used the molecular weight marker to determine the mass (original blot images were attached in the supplementary file).
2.6. Authors state that there is increased phosphorylated protein in the tumor lysate. Total protein is also enhanced and it seems that there was no normalization analysis performed. Are equal amounts of total protein analyzed?
Response: We thank the reviewer’s comment. Equal amounts of total protein were analyzed.
Usually p-protein signal is normalized to the actin detected on the same membrane; also for total protein; then, the numbers obtained are used to determined phosphorylated/total protein ratio. Here most probably p-p38 increased when SKOV3 were co-injected with MSC as compared to SKOV3 alone; however, we cannot draw the same conclusion for pGSK-3b as the total protein is also increased…
Response: We thank the reviewer’s comment. We have added the normalization value in the figure 6. Indeed, the phosphorylation of GSK was decreased after adjusting with GSK total protein. However, the large amount of pGSK still could promote SKOV3 growth. The statement reads as”Although the decreased relative quantification of GSK-3β phosphorylation was noted, the total amount of GSK-3β phosphorylation was still large, which may promote tumor growth.”(page 14, line 478-480)
How do they interpret the difference taking into account that the tumor contains not only cancer cells but also other type of cells that express these proteins?
Response: We thank the reviewer’s comment. Due to BMSCs and fibroblasts not forming tumors in the mice, we cannot use them for control. We may add in vitro cultured BMSCs and fibroblasts for control in the future project.
2.7. Co-culture should be compared to single culture and cytokine secretion normalized first to number of cells (or total protein content of the media) at the endpoint for each sample.
Response: We thank the reviewer’s comment. We have normalized the data with cell numbers and added this part in the method section (section 2.14). The statements read as”The value was normalized with cell numbers.” (page 7, line 317-318)
Discussion
The conclusion is not correct:
“Ovarian cancer cells (SKOV3) co-cultured with BMSCs showed increased migration, invasion, and colony formation” All these experiments were performed using conditioned media and not co-cultures.
Response: We thank the reviewer’s comment. We have rewrote this part in the abstract and discussion sections. The statement reads as” Ovarian cancer cells (SKOV3) cultured with CM from BMSCs showed increased migration, invasion, and colony formation, indicating the role of the tumor microenvironment in influencing cancer cell behavior”. (page 15, line 514)
The article should be carefully revised and rewritten with careful consideration for data handling.
Response: We thank the reviewer’s comment. We have carefully revised and rewritten the manuscript.
Given the above observations, I propose the article for major revision.
Response: We thank the reviewer’s comment. We have revised the manuscript and replied point by point. I hope the revised manuscript can fulfill the publication criteria.
Comments on the Quality of English Language
The article should be carefully rewritten for clarity, especially while explaining comparison between different experimental conditions that are not named clearly.
Response: We thank the reviewer’s comment. We sent the manuscript to MDPI’s recommended English editing company for English editing.
Reviewer 2 Report
Comments and Suggestions for Authors
In this study, the authors investigate the interactions between bone marrow stem cells (BMSCs) and ovarian cancer cells. They found that BMSCs promoted cancer cell migration, invasion, and tumorigenesis in NOD-SCID mice using one ovarian cancer cell line, SKOV3. Co-culture of SKOV3 cells with BMSCs also led to increased phosphorylation of p38MAPK and GSK-3β in cancer cells, along with elevated expression of cytokines like MCP-1 and IL-6. These findings underscore the influence of BMSCs on ovarian cancer behavior and suggest potential therapeutic targets for halting cancer progression.
I have some concerns regarding the clarity and depth of the manuscript. Firstly, the aim of the study should be explicitly stated to help readers better understand its purpose. Additionally, while exploring the connection between cytokine production by BMSCs and its impact on tumor cell behavior in vitro is logical, the investigation's depth in xenograft experiments seems lacking rational and requires more thorough examination. This is also a critical point in the xenografts experiments as the text is not clear if the experiment was controlled for the final number of cells in the different conditions.
Moreover, relying solely on one ovarian cancer cell line throughout the manuscript raises concerns, particularly the potential of a misclassification of the SKOV3 cell line as multiple studies reported that SKOV3 is TP53 null cell line cause by a frameshift mutation. It's crucial to address this issue by confirming the authenticity of the SKOV3 cell line using appropriate methods.
Minor revisions are also necessary, such as clarifying the terminology used in the "Proliferation Rates of Ovarian Cancer Cells after Co-culture with BMSCs" on the figure 2 “Percentage of day 0” section and providing staining confirmation of BMSC and fibroblast presence in histological sections of xenograft tumors.
Furthermore, a crucial control is missing in the experiment investigating the phosphorylation of p38MAPK and GSK-3β in xenografts. Controls for BMSC and fibroblast alone are essential to ensure the observed signals are not influenced by the presence of these cells.
Lastly, the introduction predominantly focuses on mesenchymal stem cells (MSCs), with minimal mention of bone marrow stem cells. A more detailed description of bone marrow stem cells would enhance the clarity of the introduction.
Comments on the Quality of English LanguageSome of the sentence needs to be revised.
Author Response
Reviewer 2
In this study, the authors investigate the interactions between bone marrow stem cells (BMSCs) and ovarian cancer cells. They found that BMSCs promoted cancer cell migration, invasion, and tumorigenesis in NOD-SCID mice using one ovarian cancer cell line, SKOV3. Co-culture of SKOV3 cells with BMSCs also led to increased phosphorylation of p38MAPK and GSK-3β in cancer cells, along with elevated expression of cytokines like MCP-1 and IL-6. These findings underscore the influence of BMSCs on ovarian cancer behavior and suggest potential therapeutic targets for halting cancer progression.
I have some concerns regarding the clarity and depth of the manuscript. Firstly, the aim of the study should be explicitly stated to help readers better understand its purpose. Additionally, while exploring the connection between cytokine production by BMSCs and its impact on tumor cell behavior in vitro is logical, the investigation's depth in xenograft experiments seems lacking rational and requires more thorough examination. This is also a critical point in the xenografts experiments as the text is not clear if the experiment was controlled for the final number of cells in the different conditions.
Response: We thank the reviewer’s comment. We have added the rationale for doing xenograft experiments (section 2.10). We have added a more detailed experiment protocol in the method (section 2.10). The statements read as:
“To know the role of BMSCs and fibroblasts in promoting tumor growth, xenograft experiment (SKOV3 alone, SKOV3+BMSC, SKOV3+FB02) was performed.” (page 6, line 254-256)
“Tumor cells (SKOV3, unknown passage, 1 × 10^5 cells) alone or in combination with BMSCs (P7, 3 × 10^5 cells) or fibroblasts (P6, 3 × 10^5 cells) (n = 3 each) were subcutaneously injected into the back area of the mice.” (page 6, lines 266-268)
Moreover, relying solely on one ovarian cancer cell line throughout the manuscript raises concerns, particularly the potential of a misclassification of the SKOV3 cell line as multiple studies reported that SKOV3 is TP53 null cell line cause by a frameshift mutation. It's crucial to address this issue by confirming the authenticity of the SKOV3 cell line using appropriate methods.
Response: We thank the reviewer’s comment. SKOV3 cells are a gift from another lab. They marked the cells as SKOV3. We have sent the cell line to authentication. They need 2-3 weeks for the process. Till we reply, the process is still ongoing. We performed qPCR and showed that the expression of TP53 was lower in SKOV3 than in the primary HGSC cells (p53 overexpression and p53 lower expression). We upload the picture as a supplementary file.
Minor revisions are also necessary, such as clarifying the terminology used in the "Proliferation Rates of Ovarian Cancer Cells after Co-culture with BMSCs" on the figure 2 “Percentage of day 0” section and providing staining confirmation of BMSC and fibroblast presence in histological sections of xenograft tumors.
Response: We thank the reviewer’s comment. In Figure 2, we have changed the “percent of day 0” to “O.D.” of the cell (checked by XTT, which means cell numbers). We have done the IHC of CD73 (an MSC marker) and α-SMA (a fibroblast marker) experiment in xenograft tumors. However, we did not find CD73 in the tumor mixed with BMSCs. α-SMA was expressed in all three tumor tissues. This means no BMSCs existed in xenograft after 39 days of in vivo tumor proliferation. α-SMA indicates a myofibroblast-like cell marker and an indicator of microvascularization within the tumor (Spaeth et al. 2009). a-SMA may not be a specific marker for fibroblasts. We prefer not to show the data.
Spaeth et al. Mesenchymal stem cell transition to tumor-associated fibroblasts contributes to fibrovascular network expansion and tumor progression. PLoS One. 2009;4(4):e4992.
Furthermore, a crucial control is missing in the experiment investigating the phosphorylation of p38MAPK and GSK-3β in xenografts. Controls for BMSC and fibroblast alone are essential to ensure the observed signals are not influenced by the presence of these cells.
Response: We thank the reviewer’s comment. We agree with the reviewer’s comment. However, BMSC and fibroblast cannot form tumors in vivo. We may use in vitro culture BMSC and fibroblast as controls in future projects.
Lastly, the introduction predominantly focuses on mesenchymal stem cells (MSCs), with minimal mention of bone marrow stem cells. A more detailed description of bone marrow stem cells would enhance the clarity of the introduction.
Response: We thank the reviewer’s comment. We have added a paragraph introducing BMSC. The statements read as” BMSCs are multipotent cells found within the bone marrow that have the capacity to differentiate into various cell types, including blood cells (such as red blood cells, white blood cells, and platelets) and cells of the mesenchymal lineage (such as osteoblasts, adipocytes, and chondrocytes) [5]. These cells play a crucial role in replenishing and maintaining the body's blood cell supply through hematopoiesis, as well as in tissue repair and regeneration through their ability to differentiate into specialized cell types [6]. BMSCs have garnered significant attention in research and clinical applications due to their regenerative potential and ability to contribute to the treatment of various diseases and injuries [7]. BMSCs produce cytokines like interleukins (IL-6, IL-8), tumor necrosis factor-alpha (TNF-α), and transforming growth factor-beta (TGF-β), influencing self-renewal, differentiation, and interactions within the bone marrow niche [8]. Additionally, BMSCs respond to cytokines from neighboring cells; inflammatory cytokines stimulate anti-inflammatory responses and tissue repair [9]. Hematopoietic cell-derived cytokines, like granulocyte colony-stimulating factor (G-CSF), also affect BMSC function and hematopoietic stem cell maintenance [10]. This intricate interplay regulates tissue homeostasis, regeneration, and immune modulation.“ (page 2, lines 46-61).
Comments on the Quality of English Language
Some of the sentence needs to be revised.
Response: We thank the reviewer’s comment. We sent the manuscript to MDPI’s recommended English editing company for English editing.

Reviewer 3 Report
Comments and Suggestions for Authors
The manuscript presents an interesting work about the cell proliferation promoting effect of bone marrow stem cells to ovarian tumor cells. The manuscript is well-written and contains valuable results. However, in some cases the presented information is not detailed enough. For this reason, some modifications are required to increase the value of the manuscript.
My major concerns:
- The authors present that co-culturing BMSC and ovarian cancer cells promote cancer cell proliferation. However, the biological relevance of this co-existence is not clear. How and where these cells can interact in the body? Detail this phenomenon in the introduction section.
- Add more details about the experimental design in the results section. E.g. how was proliferation/migration studied? How co-culture was carried out? Why were that surface markers studied? Why the potential of chondrogenesis, osteogenesis, adipogenesis were studied? Why were fibroblast cell cultures used as controls?
- It is not clear what Figure 2 presents in the y axis. What is percent day 0? Give a proliferation rate instead.
- The methods are not detailed enough. When not a commercially available kit was applied give a reference for the applied method/assay (e.g. in the case of migration, invasion, adipogenesis, chondrogenesis, osteogenesis…. assays).
- The clinical details about the donors of the BMSC cells is missing. Authors present only “from patients who had undergone orthopedic surgery owing to trauma”. E.g.: How many patients? In what age? What kind of trauma? Do they have any diseases?
- Details about cell culturing is also missing. What was the composition of the culturing medium in the case of the SKOV3 cultures? E.g. FBS concentration? It was also not added how old the cultures were in the time of the experiments.
- More details are required in the case of the co-culture. What were the steps of co-culturing? How were the cells plated? How old the culture was in the beginning of co-culturing? How long the experiment was?
- Some details about the preparation of the animal studies are missing too. How the cells were cultured before injection (e.g. in what medium, for how many days, how old cultures were used). What mixture of cells were injected to the mice? What was the duration of the experiment when the mice were sacrificed after injection? The authors added that “The mice were sacrificed when the tumor reached 500 mm3 in size”. However, according to Figure 5 the tumors were larger in the case of the BMSC-SKOV3 cells. How was the tumor size scanned in the living mice? The ethical approval about animal studies is missing.
- Authors concluded that the growth promoting effect of BMSC-SKOV3 co-cultures was the consequence of elevated cytokine production. What was the cytokine production of BMSC only cultures (Figure 7)? Have you studied the effect of purified cytokines added to SKOV3 cultures?
- In the discussion section add more references about other studies using co-cultures.
- The conclusion is too short. Explain the clinical relevance of your study. E.g.: future directions, use in therapy or diagnostics…
Comments on the Quality of English LanguageMinor editing of english is required.
Author Response
Reviewer 3
The manuscript presents an interesting work about the cell proliferation promoting effect of bone marrow stem cells to ovarian tumor cells. The manuscript is well-written and contains valuable results. However, in some cases the presented information is not detailed enough. For this reason, some modifications are required to increase the value of the manuscript.
My major concerns:
- The authors present that co-culturing BMSC and ovarian cancer cells promote cancer cell proliferation. However, the biological relevance of this co-existence is not clear. How and where these cells can interact in the body? Detail this phenomenon in the introduction section.
Response: We thank the reviewer’s comment. We have added a paragraph introducing how and where these cells can interact.
The statements read as”Within the primary ovarian tumor microenvironment, BMSCs may migrate to the tumor stroma in response to chemotactic signals released by cancer cells or other stromal cells [11]. Once in the tumor vicinity, BMSCs can directly interact with ovarian cancer cells through cell-to-cell contact or paracrine signaling [12,13]. As ovarian cancer progresses, metastases often occur in the peritoneal cavity, including the omentum, peritoneum, and abdominal organs [14]. BMSCs may home in on these metastatic sites and contribute to the formation of a supportive microenvironment for cancer cell survival and growth [15]. The bone marrow serves as a reservoir for stem and progenitor cells, including BMSCs. Ovarian cancer cells may secrete factors that recruit BMSCs to the bone marrow niche, where they could potentially influence cancer cell behavior and contribute to the establishment of bone metastases [16]. The interaction between BMSCs and ovarian cancer cells can occur through various mechanisms, including paracrine signaling, exosome-mediated communication, and direct cell-to-cell contact [17–19]. “ (page 2, lines 62-74).
- Add more details about the experimental design in the results section. E.g. how was proliferation/migration studied? How co-culture was carried out? Why were that surface markers studied? Why the potential of chondrogenesis, osteogenesis, adipogenesis were studied? Why were fibroblast cell cultures used as controls?
Response: We thank the reviewer’s comment. We have added more details about the experimental design in the results section.
. E.g. how was proliferation/migration studied? Section 2.5, 2.7
How co-culture was carried out? Section 2.14. The statements read as”Corning transwell plates (pore size 8.0 μm; Corning, NY, USA) were used for the co-culture experiments. The upper chamber contained 5 × 104 BMSCs or FB02 in 0.3 mL of a culture medium, while the bottom chamber contained 1 × 105 SKOV3 cells in 0.6 mL of a culture medium and co-cultured for 24 hrs. Fibroblasts were used as the controls.” (page 7, lines 307-311)
Why were that surface markers studied? Section 2.4. The statement reads as”These markers are typical surfacer markers for BMSCs.” (page 4, line 156)
Why the potential of chondrogenesis, osteogenesis, adipogenesis were studied? Section 2.6. The statement read as”Tri-lineage differentiation (adipogenesis, osteogenesis, and chondrogenesis) is a typical characteristic of MSCs [32].” (page 4, line 176-177)
Why were fibroblast cell cultures used as controls? Section 2.2. The statements read as” Fibroblasts were used as controls in the experiment involving the interaction of BMSCs with SKOV3 tumor cells to provide a baseline comparison for assessing the specific effects of BMSCs. Fibroblasts, as a common stromal cell type found in the tumor microenvironment [28], serve as a relevant control with which to evaluate the impact of BMSCs on tumor proliferation independent of generic stromal cell effects.”
- It is not clear what Figure 2 presents in the y axis. What is percent day 0? Give a proliferation rate instead.
Response: In response to the reviewer’s comment, we have changed Figure 2 y-axis from the “percent of Day 0: to the “O.D. (optical density)” value of the cells evaluated with XTT assay. (Section 2.5. page 4, lines 169-173)
- The methods are not detailed enough. When not a commercially available kit was applied give a reference for the applied method/assay (e.g. in the case of migration, invasion, adipogenesis, chondrogenesis, osteogenesis…. assays).
Response: We thank the reviewer’s comment. We have added more information regarding the commercial kit and reference for the applied method/assay.
migration: Section 2.7 (page 5)
invasion: Section 2.8 (page 5)
adipogenesis, chondrogenesis, osteogenesis…: Section 2.6 (page 4)
- The clinical details about the donors of the BMSC cells is missing. Authors present only “from patients who had undergone orthopedic surgery owing to trauma”. E.g.: How many patients? In what age? What kind of trauma? Do they have any diseases?
Response: We thank the reviewer’s comment. We have added the patient characteristics in the section 2.1. The statement reads as” The cells were obtained from a female patient, aged 76, who suffered from a left femoral neck fracture and received bipolar hemiarthroplasty surgery”. (section 2.1, page 3, line 105-107)
- Details about cell culturing is also missing. What was the composition of the culturing medium in the case of the SKOV3 cultures? E.g. FBS concentration? It was also not added how old the cultures were in the time of the experiments.
Response: We thank the reviewer’s comment. We have added the culture medium composition in the section. 2.3. The concentration of FBS was 10%. The passage of SKOV3 cells in our lab was not known. (page 3, lines 143-144)
- More details are required in the case of the co-culture. What were the steps of co-culturing? How were the cells plated? How old the culture was in the beginning of co-culturing? How long the experiment was?
Response: We thank the reviewer’s comment. We have added the description the detailed method of each experiment. The BMSCs condition medium was added to the dish-cultured SKOV3 cells in proliferation and colony-forming assays. In the migration and invasion assay, the BMSCs condition medium was added to a transwell's lower well, and SKOV3 cells were plated in the upper well. The original co-culturing section was deleted.
In section 2.14: The statements read as”Corning transwell plates (pore size 8.0 μm; Corning, NY, USA) were used for the co-culture experiments. The upper chamber contained 5 × 10^4 BMSCs or FB02 in 0.3 mL of a culture medium, while the bottom chamber contained 1 × 10^5 SKOV3 cells in 0.6 mL of a culture medium and co-cultured for 24 hrs. Fibroblasts were used as the controls.” (page 7, line 307-311)
- Some details about the preparation of the animal studies are missing too. How the cells were cultured before injection (e.g. in what medium, for how many days, how old cultures were used). What mixture of cells were injected to the mice? What was the duration of the experiment when the mice were sacrificed after injection? The authors added that “The mice were sacrificed when the tumor reached 500 mm3 in size”. However, according to Figure 5 the tumors were larger in the case of the BMSC-SKOV3 cells. How was the tumor size scanned in the living mice? The ethical approval about animal studies is missing.
Response: We thank the reviewer’s comment. We have answered the question and added it in section 2.10.
in what medium: respected culture medium (line 270)
for how many days? Ans: 7 days (line 269)
how old cultures were used: unknown passage for SKOV3, p7 for BMSCs, p6 for fibroblast (line 276)
What mixture of cells was injected into the mice? Matrigel and culture medium:(1:1 in volume, 100 μL each) (line 270)
What was the duration of the experiment when the mice were sacrificed after injection?
Ans: 39 days. (line 277)
The authors added that “The mice were sacrificed when the tumor reached 500 mm3 in size”. However, according to Figure 5, the tumors were larger in the case of the BMSC-SKOV3 cells. How was the tumor size scanned in the living mice?
Ans: Tumor volume was recorded weekly using the following equation: volume = (width)2 ⋅ length/2 (Wang et al. 2019). (line 276)
The ethical approval about animal studies is missing.
Ans: The Animal Research and Care Committee of Hualien Tzu Chi Hospital approved the animal experiments (No. 111-31). (page 6, line 258)
- Authors concluded that the growth promoting effect of BMSC-SKOV3 co-cultures was the consequence of elevated cytokine production. What was the cytokine production of BMSC only cultures (Figure 7)? Have you studied the effect of purified cytokines added to SKOV3 cultures?
Response: We have done the suggested experiments in response to the reviewer's comment (Figure 2b). After adding IL-6, MCP01, or a combination, they all promoted SKOV3 cell proliferation.
- In the discussion section add more references about other studies using co-cultures.
Response: We thank the reviewer’s comment. We have added a paragraph to add more references about the studies using coculture experiments. The statements read as” The effects of mesenchymal stem cells (MSCs) co-cultured with ovarian cancer cells exhibit various outcomes. On the one hand, MSCs have been shown to enhance the malignant characteristics of ovarian cancer. For instance, the co-culturing of ovarian cancer stem-like cells with macrophages induced increased stemness in SKOV3 cells via IL-8/STAT3 signaling [53]. Additionally, when BMSCs were co-cultured with ovarian cancer cells, they induced a pro-metastatic transcriptomic profile, albeit dependent on the cellular context [54]. Moreover, human umbilical cord MSCs have been found to stimulate ovarian tumor growth during intercellular communication, although they exhibited reduced tumorigenicity after fusion with ovarian cancer cells [19]. Similarly, human omental-derived adipose stem cells were reported to increase ovarian cancer proliferation, migration, and chemoresistance [55]. In vitro, SKOV3 tumor cell proliferation is driven by tumor-stimulated MSC-secreted IL6, leading to the emergence of a tumor-associated fibroblast phenotype, ultimately facilitating tumor growth via microvascularization, stromal networks, and the production of tumor-stimulating paracrine factors [56]. In the current study, our findings align with the former group, indicating that BMSCs induce malignant behavior in ovarian cancer cells. This highlights the diverse and context-dependent interactions between MSCs and ovarian cancer cells, emphasizing the need for further investigation to elucidate the underlying mechanisms and potential therapeutic implications. “(page 16, lines 547-565).
- The conclusion is too short. Explain the clinical relevance of your study. E.g.: future directions, use in therapy or diagnostics…
Response: We are grateful for the reviewer’s comment. In response to the comment, we have added the conclusion regarding future direction, and use in therapy and diagnostics. The statements read as”Understanding the mechanisms underlying the BMSC-mediated promotion of ovarian cancer progression opens up avenues for developing novel therapeutic strategies. Targeting specific cytokines or signaling pathways identified in our study, such as MCP-1, IL-6, p38MAPK, and GSK-3β, could potentially disrupt the tumor-promoting effects of BMSCs and inhibit ovarian cancer progression. The identified cytokines and signaling pathways could serve as biomarkers for assessing ovarian cancer progression and predicting patient outcomes. Monitoring the expression levels of MCP-1 and IL-6, or the activation status of p38MAPK and GSK-3β, in tumor tissues or patient serum may provide valuable diagnostic information and aid in treatment decision making. Further research is warranted to elucidate the precise mechanisms underlying the crosstalk between BMSCs and ovarian cancer cells. Investigating the additional cytokines, signaling pathways, and molecular interactions involved in this interplay could uncover new therapeutic targets and refine existing treatment strategies for ovarian cancer.” (page 16, lines 576-588)
Comments on the Quality of English Language
Minor editing of english is required.
Response: We thank the reviewer’s comment. We sent the manuscript to MDPI’s recommended English editing company for English editing.
Round 2
Reviewer 1 Report
Comments and Suggestions for Authors
The article has been improved by additional controls added to figures and methods subsections that were previously lacking.
However, there are several aspects that jeopardize the scientific soundness of the work if left unclarified/uncorrected.
The following issues attract the attention and have to be corrected:
Please remove sentence related to ref 16 if not backed up by other studies. It is not clear how cells in the peritoneum might attract cells into BM…
Also, 100000 cells do not normally fit into a 12-well, especially MSC, which are very large cells; moreover, MSC harvested from an old individual cannot differentiate into osteoblasts in 14 days. Usually it takes 21-28 days for this type of differentiation protocol, even with young cells. The reference cited refers to pre-conditioned MSCs, which are already stimulated and not plain MSCs.
The IHC method is incompletely described, which hinders reproducibility. Text should describe slides processing conditions (paraffin removal, rehydration, epitope retrieval etc. ) and not only the last steps of staining.
The ELISA should not generate any measurable signal with the method described. Adding streptavidin-HRP on top of analyte captured with plate coated antibodies would not bind…Please correct the sandwich ELISA description or repeat the experiment. The ELISA kit catalog number is also missing.
Figure 3: Most probably the images presented are taken with the 10x objective, based on relative cells size and scale bar (100x magnification). It is not clear why authors say the cells are not visible at 40x magnification but this is the magnification they used to count them…
Figure 4: The graph still presents “precent” instead of “number of colonies formed”. If authors want to report percent of colonies occupied area of total area of the well, this value cannot possibly be 100% for SKOV3 and 300% for SKOV3 grown in BMSC media. Also, how can they explain the lack of colonies in the presence of FB02 CM? That would mean that the factors secreted by fibroblasts are anti-tumorigenic?! The experiment should be repeated.
In order to perform a correct assessment of percentage occupied surface, one should quantify the purple area with Image J and normalize it to the 6-well area for each well of the experiment and then report the values as “percent occupied surface”.
Western blot representation and data analysis is one of the most worrying issues. While authors have added protein content determination description, it is still not clear how much protein was added to the SDS-PAGE gel from each sample or if there were equal quantities loaded.
It is very hard to draw any conclusion from the phosphoprotein expression analysis. Although a normalization has been made and numbers appear in the figure, it is not described in materials and methods how it was performed. Also, the explanation in figure caption is not clear. How can the highly intense pGSK-3b band in SKOV3+BMSC be considered 0.36 and the total GSK-3b (similarly intense) be 14.69?!
Also, cropped image of WT1 protein expression is not convincing due to elimination of an upper band much more intense than the one selected.
Cytokine array: authors state in methodology that they retained media from BMSCs and FB02. However, they do not show any results from baseline secretion of these cell types in Fig 7 A and B.
Also, the methodology describes FB02 media as control, while there is no result of this cell line in Fig 7 C and D.
These inconsistencies have to be clarified before the article is acceptable for publication.
Discussion: Authors seem to enumerate a series of studies but the order of ideas does not follow a logic sequence. While describing the known reports on MSCs influence on ovarian cancer cells, they report on a study using macrophages (ref 53) and then come back to MSCs related studies…
Minor details:
-page 5, line 232 – the colonies were fixed via fixedwith…
-page 6, line 279 – the software used for spot densitometry should be mentioned
-page 13, lines 438, 441, 448 – “expression” should be replaced with “secretion”
Given the above observations, the article still needs major revision.
Comments on the Quality of English LanguageEnglish language still needs moderate editing to provide clarity to the flow of ideas.
Author Response
Reviewer 1
Comments and Suggestions for Authors
The article has been improved by additional controls added to figures and methods subsections that were previously lacking.
However, there are several aspects that jeopardize the scientific soundness of the work if left unclarified/uncorrected.
The following issues attract the attention and have to be corrected:
Please remove sentence related to ref 16 if not backed up by other studies. It is not clear how cells in the peritoneum might attract cells into BM…
Response: We thank the reviewer’s comment. We have removed the sentence related to ref 16 (previous version line 67-69).
Also, 100000 cells do not normally fit into a 12-well, especially MSC, which are very large cells; moreover, MSC harvested from an old individual cannot differentiate into osteoblasts in 14 days. Usually it takes 21-28 days for this type of differentiation protocol, even with young cells. The reference cited refers to pre-conditioned MSCs, which are already stimulated and not plain MSCs.
Response: We thank the reviewer’s comment. Our previous study used 1 x 10^4 cells differentiated for 14 days. Nevertheless, in this study, we used 1 x 10^5 cells differentiated for 14 days. The Alizarin red staining showed differentiated cells could be stained, revealing the typical osteoblast characteristic. We referenced our previous study (Chang et al. 2022). (page 4, line 180)
Chang, Yu-Hsun, V. Bharath Kumar, Yao-Tseng Wen, Chih-Yang Huang, Rong-Kung Tsai, and Dah-Ching Ding. 2022. “Induction of Human Umbilical Mesenchymal Stem Cell Differentiation Into Retinal Pigment Epithelial Cells Using a Transwell-Based Co-Culture System.” Cell Transplantation 31: 9636897221085901.
The IHC method is incompletely described, which hinders reproducibility. Text should describe slides processing conditions (paraffin removal, rehydration, epitope retrieval etc. ) and not only the last steps of staining.
Response: We thank the reviewer’s comment. We added the detailed protocol of IHC, including paraffin removal, rehydration, and epitope retrieval (section 2.12). (page 6, lines 257-286)
The statements read as”For deparaffinization and rehydration, place the slides in a 56-60°C oven for 15 minutes. Transfer the slides to a xylene bath and perform two changes of xylene, each for 5 minutes. Shake off the excess liquid and rehydrate the slides in two changes of fresh absolute ethanol, each for 5 minutes. Next, shake off the excess liquid and place the slides in fresh 90% ethanol for 5 minutes, followed by fresh 80% ethanol for 5 minutes. Rinse the slides under gently running tap water for 30 seconds, avoiding a direct jet to prevent washing off or loosening the section. Finally, place the slides in a PBS wash bath for further rehydration for 5 minutes at room temperature.
Enzyme retrieval was performed as follows. Apply 0.1% trypsin in PBS for 2-30 minutes at 37°C. After incubation, rinse the slides in PBS for 10 minutes. Soaked slides with sodium citrate buffer working solution and incubated for 15 minutes at 114-120 °C.
Add 3% hydrogen peroxide to cover the entire section and incubate for 10 minutes at room temperature. After incubation, rinse the slide with PBS using a wash bottle, then place it in a PBS wash bath for 2 minutes. Ultra V blocked for 5 minutes.
Pre-incubated the sample with 5% BSA (bovine serum albumin, Sigma) for 10 minutes before the primary antibody reaction.
Anti-P53 (1:100, 2527, rabbit mAb, Cell Signaling) and WT1 (1:100, BSB 6033, Bio SB) monoclonal antibodies were used for IHC. The sections were incubated with primary antibodies overnight, then PBS washed for 5 minutes three times. Then, the primary antibody amplifier Quanto (Thermo Scientific) was added for 10 minutes. PBS washed for 5 minutes three times. Adding horseradish peroxidase (HRP) polymer Quanto (Thermo Scientific) for 10 minutes. PBS washed for 5 minutes three times. Apply diaminobenzidine tetrahydrochloride (DAB) Quanto Chromogen and DAB Quanto substrate mix to tissue for 5 mintuers. Distilled water wash for 5 minutes three times. Add hematoxylin for 5 minutes and rinse with distilled water for 5 minutes three times.
Photographs of the stained sections were recorded using a light microscope (Nikon TE2000-U fitted with a digital camera [Nikon DXM1200F], Nikon, Tokyo, Japan). The positive-stained cells (brown color in the nucleus) of 100 cells in the 5 randomly selected fields were calculated.”
The ELISA should not generate any measurable signal with the method described. Adding streptavidin-HRP on top of analyte captured with plate coated antibodies would not bind…Please correct the sandwich ELISA description or repeat the experiment. The ELISA kit catalog number is also missing.
Response: We thank the reviewer’s comment. We have added the detailed ELISA method and ELISA kit catalog number. (Section 2.14) The statements read as”The concentrations of cytokines in the medium were quantified using ELISA kits for IL-6 (#BMS213-2, Invitrogen) and MCP-1 (#BMS281, Invitrogen). Corning transwell plates (pore size 8.0 μm; Corning, NY, USA) were used for the co-culture experiments. The upper chamber contained 5 × 104 BMSCs or FB02 in 0.3 mL of a culture medium, while the bottom chamber contained 1 × 105 SKOV3 cells in 0.6 mL of a culture medium and co-cultured for 24 hrs. Fibroblasts were used as the controls. The medium derived from BMSCs or BMSCs co-cultured with SKOV3 was incubated on an antibody-coated plate for one hour. After washing, the plates were incubated with biotin-conjugated secondary antibody for 2 hours, streptavidin–horseradish peroxidase (HRP, Abcam) for 30 minutes, and chromogen for 10 minutes. After the reaction was stopped, the plates were analyzed using an ELISA reader at 450 nm as the primary wavelength and 620 nm as the reference wavelength. The difference between OD450 and OD620 and the blank were fitted into the standard curve to determine the target protein concentration. The value was normalized with cell numbers.”
Figure 3: Most probably the images presented are taken with the 10x objective, based on relative cells size and scale bar (100x magnification). It is not clear why authors say the cells are not visible at 40x magnification but this is the magnification they used to count them…
Response: We thank the reviewer’s comment. We changed the pictures to 40x magnification (Figure 3). (page 11)
Figure 4: The graph still presents “precent” instead of “number of colonies formed”. If authors want to report percent of colonies occupied area of total area of the well, this value cannot possibly be 100% for SKOV3 and 300% for SKOV3 grown in BMSC media. Also, how can they explain the lack of colonies in the presence of FB02 CM? That would mean that the factors secreted by fibroblasts are anti-tumorigenic?! The experiment should be repeated.
In order to perform a correct assessment of percentage occupied surface, one should quantify the purple area with Image J and normalize it to the 6-well area for each well of the experiment and then report the values as “percent occupied surface”.
Response: We thank the reviewer’s comment. We replaced the CFU experiments with an anchorage-independent growth assay, representing the in vitro tumorigenesis ability (Figure 4).(page 12).
The related methods were described in the section 2.9 (page 5, lines 221-228)
Western blot representation and data analysis is one of the most worrying issues. While authors have added protein content determination description, it is still not clear how much protein was added to the SDS-PAGE gel from each sample or if there were equal quantities loaded.
Response: We thank the reviewer’s comment. We loaded 20 μg of protein from each sample to perform a Western blot. We added the protein amount in the method section (section 2.15). (page 7, line 320)
It is very hard to draw any conclusion from the phosphoprotein expression analysis. Although a normalization has been made and numbers appear in the figure, it is not described in materials and methods how it was performed. Also, the explanation in figure caption is not clear. How can the highly intense pGSK-3b band in SKOV3+BMSC be considered 0.36 and the total GSK-3b (similarly intense) be 14.69?!
Response: We thank the reviewer’s comment. We redid the experiment (new Figure 6). The ratio of each protein was recalculated. The results showed phosphorylation of p38 MAPK and GSK-3β increased in SKOV3 co-cultured with BMSCs and FB02. (page 14)
Also, cropped image of WT1 protein expression is not convincing due to elimination of an upper band much more intense than the one selected.
Response: We thank the reviewer’s comment. We redid the experiment (new Figure 5). The results were the same as the previous results. (page 13)
Cytokine array: authors state in methodology that they retained media from BMSCs and FB02. However, they do not show any results from baseline secretion of these cell types in Fig 7 A and B.
Also, the methodology describes FB02 media as control, while there is no result of this cell line in Fig 7 C and D.
Response: We thank the reviewer’s comment. We have added the cytokine array data of BMSC and FB02 in Figure 7 A (page 16). We also added the results of the ELISA of the FB02 condition medium to the new Figures 8C and D. (page 17)
These inconsistencies have to be clarified before the article is acceptable for publication.
Response: We thank the reviewer’s comment. We have resolved the inconsistency issue and hope this manuscript can be accepted for publication.
Discussion: Authors seem to enumerate a series of studies but the order of ideas does not follow a logic sequence. While describing the known reports on MSCs influence on ovarian cancer cells, they report on a study using macrophages (ref 53) and then come back to MSCs related studies…
Response: We thank the reviewer’s comment. We have rewritten the paragraph in a logical manner and deleted ref 53. (page 18, lines 524-543)
Minor details:
-page 5, line 232 – the colonies were fixed via fixedwith…
Response: The paragraph was replaced by new methods (Section 2.9).
-page 6, line 279 – the software used for spot densitometry should be mentioned
Response: We have added the software used for spot densitometry. The statement reads as”The cytokines were quantified by spot densitometry with ImageJ software (NIH).” (page 7, line 294)
-page 13, lines 438, 441, 448 – “expression” should be replaced with “secretion”
Response: We have replaced “expression” with “secretion”. (page 15. line 462 and 470)
Given the above observations, the article still needs major revision.
Comments on the Quality of English Language
English language still needs moderate editing to provide clarity to the flow of ideas.
Response: We have sent the manuscript for English editing.
Reviewer 2 Report
Comments and Suggestions for Authors
The authors have addressed some of the points raised in the previous round of revisions. However, there are still major issues that need to be resolved, which are crucial for drawing conclusions from the experiments. Merely stating that the authentication of the cell line (a unique ovarian tumor cell line set throughout the manuscript) is being performed is not adequate. From the reviewer's standpoint, this confirmation is essential and should be provided. Another concern is the lack of proper controls in the western blot experiments from the xenograft studies, particularly regarding the p38 MAPK and GSK-3B signaling pathways. Simply stating that "We may use in vitro culture BMSC and fibroblast as controls in future projects" is insufficient. The conclusions drawn should refer to this manuscript and not rely on potential future projects.
Comments on the Quality of English LanguageMinor points were adressed
Author Response
Reviewer 2
Comments and Suggestions for Authors
The authors have addressed some of the points raised in the previous round of revisions. However, there are still major issues that need to be resolved, which are crucial for drawing conclusions from the experiments. Merely stating that the authentication of the cell line (a unique ovarian tumor cell line set throughout the manuscript) is being performed is not adequate. From the reviewer's standpoint, this confirmation is essential and should be provided. Another concern is the lack of proper controls in the western blot experiments from the xenograft studies, particularly regarding the p38 MAPK and GSK-3B signaling pathways. Simply stating that "We may use in vitro culture BMSC and fibroblast as controls in future projects" is insufficient. The conclusions drawn should refer to this manuscript and not rely on potential future projects.
Response: We thank the reviewer’s comment. We have provided the authentication of SKOV3. The cell line was 100% identical to the SKOV3 STR analysis (the attachment file for the reviewer).
We have done the protein expression of p38 MAPK and GSK-3B signaling pathways of BMSC and fibroblast in new Figure 6. (page 14)
Comments on the Quality of English Language
Minor points were adressed

Reviewer 3 Report
Comments and Suggestions for Authors
The authors responded my concerns and made several modifications in the manuscript.
Author Response
Reviewer 3
Comments and Suggestions for Authors
The authors responded my concerns and made several modifications in the manuscript.
Response: We thank the reviewer’s comment.
Round 3
Reviewer 1 Report
Comments and Suggestions for Authors
The article has been improved by additional experiments replacing previously presented results that were not convincing.
However, there are several aspects that still needs correction.
Regarding the newly added text (methodology/results), there are several errors that need to be corrected:
The paragraph describing the IHC method (section 2.12) is a copy/pasted protocol that needs to be rewritten as a description, not as indications to the reader?!?
Also, the newly performed anchorage-independent growth assay (section 2.9) is described by mixing protocol text with method description. The sentence “Feed cells two times a week” should be changed into “Media was replaced two times weekly.”
Regarding the phosphoprotein expression analysis by Western blotting (sections 2.15 and 3.6), there are a few aspects that still need to be considered:
-A phrase should be added at the beginning of 3.6 to explain why pGSK3-beta and p-p38 were specifically analyzed in this context. In conclusions, authors state they identified new signaling pathways involved in ovarian cancer tumorigenesis that could be targetable. How were this identified/selected? Either indicate data or literature to support this selection.
-The signals for the phospho-proteins are too low, especially for pGSK3-beta. A few potential causes are use of 1:2000 dilution of antibodies instead of the Cell Signaling recommended 1:1000 dilution and use of ECL instead of ECL Femto for bands detection.
-The methods section still fail to describe the formula for normalization. Each investigated protein should be normalized to the loading control (actin??? The bands do not show equal loading) detected on the same membrane; then, the ratio of the values obtained for the p-protein and the total protein should be calculated to evaluate activation/inhibition. Also, a good practice is to include an activator/inhibitor as control to treat cells for signaling modulation (e.g. PDGF for p-GSK3-beta, as indicated on the Cell Signaling website).
Here, all the proteins have similar molecular weight, so they cannot be detected simultaneously on the same membrane (p-GSK3-beta/actin). Also, both pGSK3-beta (#9323) and actin (#4967; wrongly stated #4697) are produced in rabbit, which makes impossible independent serial detection using alternatively anti-rabbit and anti-mouse antibodies.
Please correct or clarify how this analysis was performed.
Conceptually, authors use generic terms such as bone marrow stem cellsthat might confuse the readers. These are used here to describe the cancer-associated mesenchymal stem cells, as in their previous review paper (DOI: 10.4103/tcmj.tcmj_138_22) or ovarian mesenchymal stem cells, as in their experimental work (doi: 10.7150/jca.16116, cited as ref 25). Bone marrow mesenchymal stem cells (usually abbreviated BM-MSC) represent a fraction of the bone marrow stem cell population and are the progenitors for adipocytes, osteoblasts and chondrocytes (adherent cells). Another bone marrow stem cell fraction is represented by the hematopoietic stem cells, which give rise to blood immune cells (non-adherent fraction).
In the present paper, authors use alternatively “BMSC” (including in the title) to either define all the stem cells in the bone marrow (page 1, line 43) or to name the mesenchymal stem cell fraction (page 2, line 56), probably to maintain the BMSC label for the cell line they previously generated (BMSC-05).
This should be clarified throughout the text and the title, which should state: “Bone marrow mesenchymalstem cells promote ovarian cancer cell proliferation via cytokine-mediated interactions”. Also, it should be better clarified what is the novelty of this study as opposed to their previous study showing MSC secrete IL-6 thereby promoting proliferation and tumorigenesis of ovarian cancer cells (ref 25; lines 84-90).
Given the above observations, the article still needs minor revision.
Comments on the Quality of English LanguageIn section 3.4 there is again the same phrase that was corrected before during the first review: page 11, line 406: “compared to without culturing!?”
Careful check of language flow should be performed.
Author Response
as attachment file

Reviewer 2 Report
Comments and Suggestions for Authors
The authors addressed the major points raised in the previous round of revision. No additional concerns remain.
Comments on the Quality of English LanguageThe authors addressed the major points raised in the previous round of revision. No additional concerns remain.
Author Response
We thank the reviewer’s comments.